# Chemical processes related to net ozone tendencies in the free troposphere

Heiko Bozem[1*], Tim M. Butler[2], Mark G. Lawrence[2], Hartwig Harder[1], Monica Martinez[1], Dagmar Kubistin[1#], Jos Lelieveld[1] and Horst Fischer[1]

[1]Max Planck Institute for Chemistry, POB 3060, 55020 Mainz, Germany
[2]IASS Potsdam, Berliner Strasse 30, 14467 Potsdam, Germany
[*]now at Johannes-Gutenberg University, Mainz, Germany
[#]now at DWD, Hohenpeißenberg, Germany

*Correspondence to*: Heiko Bozem (bozemh@uni-mainz.de)

**Abstract.** Ozone ($O_3$) is an important atmospheric oxidant, a greenhouse gas, and a hazard to human health and agriculture. Here we describe airborne in-situ measurements and model simulations of $O_3$ and its precursors during tropical and extratropical field campaigns over South America and Europe, respectively. Using the measurements, net ozone formation/destruction tendencies are calculated and compared to 3D chemistry-transport model simulations. In general, observation-based net ozone tendencies are positive in the continental boundary layer and the upper troposphere at altitudes above ~ 6 km in both environments. On the other hand, in the marine boundary layer and the middle troposphere, from the top of the boundary layer to about 6-8 km altitude, net $O_3$ destruction prevails. The ozone tendencies are controlled by ambient concentrations of nitrogen oxides ($NO_x$). In regions with net ozone destruction the available $NO_x$ is below the threshold value at which production and destruction of $O_3$ balance. While threshold NO values increase with altitude, in the upper troposphere $NO_x$ concentrations are generally higher due to the integral effect of convective precursor transport from the boundary layer, downward transport from the stratosphere and $NO_x$ produced by lightning. Two case studies indicate that in fresh convective outflow of electrified thunderstorms net ozone production is enhanced by a factor 5 – 6 compared to the undisturbed upper tropospheric background. The chemistry-transport model MATCH-MPIC generally reproduces the pattern of observation-based net ozone tendencies, but mostly underestimates the magnitude of the net tendency (for both net ozone production and destruction).

## 1 Introduction

Ozone plays a pivotal role in the oxidising capacity of the troposphere. Besides being an oxidising agent itself, photolysis of $O_3$ at wavelengths less than 340 nm produces $O(^1D)$, whose subsequent reaction with water vapour yields two OH radicals, the dominant oxidant in the troposphere. Based on $O_3$ profiles in the troposphere, Junge (1963) argued that tropospheric ozone stems from downward transport from the stratosphere and is destroyed at the surface by deposition. But in the 1960s, studies indicated that tropospheric ozone is to a large extent due to in-situ photochemical production, similar to the Los

Angeles smog (Haagen-Smit and Fox, 1956; Leighton, 1961). A chemical mechanism for the photochemical production of tropospheric ozone was proposed by Chameides and Walker (1973) and Crutzen (1973) after the identification of a major tropospheric OH source by Levy (1971). Budget calculations based on atmospheric chemistry-transport modelling (e.g. Wild, 2007; Wu et al., 2007; Stevenson et al., 2006; von Kuhlmann et al., 2003), indicate that approximately 390-850 Tg/yr of tropospheric $O_3$ are due to stratosphere-troposphere-transport, 670-1180 Tg/yr are destroyed by deposition to the surface and -90 to +670 Tg/yr are due to photochemical net ozone production (NOP) in the troposphere (von Kuhlmann et al., 2003, Lelieveld and Dentener, 2000). The NOP itself is a delicate balance between two very large numbers (Lelieveld and Dentener, 2000; von Kuhlmann et al., 2003): ozone production $P(O_3)$ at ~3000-5000 Tg/yr and $O_3$ destruction $L(O_3)$ at slightly less, ~2500-4500 Tg/yr. The discussion about the relative strength of stratosphere-troposphere-exchange vs. NOP for tropospheric ozone is not yet resolved in detail. Gross production and destruction of ozone in a global model are based on the applied chemical mechanism, emissions of precursors and their subsequent distribution due to transport. Since stratosphere-troposphere-transport of $O_3$ depends on the gradient of $O_3$ between the lower stratosphere and upper troposphere, this process also depends on and influences photochemistry especially in the upper troposphere. Therefore, uncertainties in the models' photochemical ozone production have a strong influence on estimates of the amount of $O_3$ imported from the stratosphere. Furthermore, in chemical transport models (CTMs) and chemistry general circulation models (CGCMs) the stratospheric source of $O_3$ is often highly parameterized, e.g. with prescribed $O_3$ concentrations in the lower stratosphere to reproduce measured ozone profiles.

Tropospheric $O_3$ production is initiated by the oxidation of CO and volatile organic compounds by the OH radical:

$$CO + OH\ (+ O_2) \rightarrow HO_2 + CO_2 \tag{R1}$$

$$CH_4 + OH + (O_2) \rightarrow CH_3O_2 + H_2O \tag{R2}$$

$$RH + OH + (O_2) \rightarrow RO_2 + H_2O \tag{R3}$$

The resulting peroxy radicals $HO_2$, $CH_3O_2$ and $RO_2$ subsequently react with NO to produce $NO_2$:

$$HO_2 + NO \rightarrow NO_2 + OH \tag{R4}$$

$$CH_3O_2 + NO\ (+O_2) \rightarrow NO_2 + HO_2 + HCHO \tag{R5}$$

$$RO_2 + NO + (O_2) \rightarrow NO_2 + HO_2 + carbonyl \tag{R6}$$

Subsequently, the $NO_2$ can be photolysed to recycle NO and produce $O_3$:

$$NO_2 + h\nu\ (\lambda < 420\ nm) + (O_2) \rightarrow NO + O_3 \tag{R7}$$

In remote regions where VOC concentrations other than $CH_4$ are low the production of $O_3$ can be approximated by:

$$P(O_3) = k(4)\ [HO_2]\ [NO] + k(5)\ [CH_3O_2]\ [NO] \tag{Eq. 1}$$

With $k(4)$ and $k(5)$ being the temperature dependent rate constant of reaction R(4) and R(5). Chemical destruction of $O_3$ is either due to photolysis or reaction with OH, $HO_2$ or an alkene:

$$O_3 + h\nu\ (\lambda < 340\ nm) \rightarrow O(^1D) + O_2 \tag{R8}$$

$$O_3 + OH \rightarrow HO_2 + O_2 \tag{R9}$$

$$O_3 + HO_2 \rightarrow OH + 2\ O_2 \tag{R10}$$

$$O_3 + \text{alkene} \rightarrow \text{products} + O_2 \tag{R11}$$

Whether reaction R8 results in a permanent loss of $O_3$ depends on the fate of the electronically exited $O(^1D)$ radical. Reaction of $O(^1D)$ with either $N_2$ or $O_2$ leads to deactivation and subsequent reformation of $O_3$, but reaction with water vapour yields two OH radicals, leading to $O_3$ loss:

$$O(^1D) + H_2O \rightarrow 2\ OH \tag{R12a}$$

$$O(^1D) + O_2\ (+O_2) \rightarrow O_3 + O_2 \tag{R12b}$$

$$O(^1D) + N_2\ (+O_2) \rightarrow O_3 + N_2 \tag{R12c}$$

The branching ratio among the reactions R12a and R12b mainly depends on the water vapour concentrations and is thus altitude dependent.

In remote regions the reaction with alkenes can be neglected and the ozone loss $L(O_3)$ is given by reactions R8 – R10, with reaction R8 weighted by the branching ratio $\alpha$:

$$L(O_3) = \alpha\ JO(^1D)\ [O_3] + k(9)\ [OH][O_3] + k(10)\ [HO_2][O_3] \tag{Eq. 2}$$

Where $JO(^1D)$ is the $O_3$ photolysis rate and $\alpha$ is given by Eq. 3:

$$\alpha = \frac{k(12a)[H_2O]}{k(12a)[H_2O] + k(12b)[O_2] + k(12c)[N_2]} \tag{Eq. 3}$$

The branching ratio $\alpha$ is typically of the order of 1 to 15 % for the upper troposphere and the boundary layer, respectively.

The net ozone production rate (NOPR) in ppbv/h is defined as the difference between production and loss:

$$NOPR = P(O_3) - L(O_3) \tag{Eq. 4}$$

NOPR is nonlinear with respect to NO and peroxy radicals. This nonlinearity arises because $RO_x$ and $NO_x$ drive ozone production (R4-R6) but also terminate free radical chemistry (e.g. Pusede et al., 2015):

$$NO_2 + OH + M \rightarrow HNO_3 + M \tag{R13}$$

$$NO_2 + RO_2 + M \rightarrow NO_2RO_2 + M \tag{R14}$$

$$OH + HO_2 \rightarrow H_2O + O_2 \tag{R15}$$

$$HO_2 + HO_2 \rightarrow H_2O_2 + O_2 \tag{R16}$$

$$CH_3O_2 + HO_2 \rightarrow CH_3OOH + O_2 \tag{R17}$$

Here we neglect the loss of $NO_2$ due to reaction R13 and R14 in Eq. 4. This is justified by the overall low $NO_x$ concentrations above the continental boundary layer. Reactions R15 to R17 will affect $HO_x$ levels and would have to be taken into account to calculate their concentrations using a box model. In this study we will use in-situ observations of OH and $HO_2$ instead.

The threshold NO concentration, at which ozone production and loss are equal, can be calculated by setting

$$P(O_3) = L(O_3) \tag{Eq. 5}$$

and re-arranging for NO:

$$NO_{th} = \frac{\alpha J(O^1D)[O_3] + k(10)[HO_2][O_3] + k(9)[OH][O_3]}{k(4)[HO_2] + k(5)[CH_3O_2]} \tag{Eq. 6}$$

With reaction R4 being approximately 4000 times faster than R10 a typical range for $NO_{th}$ is 10-20 pptv at an ozone concentration of about 50 ppbv. Below this $NO_{th}$ concentration $O_3$ destruction prevails, while net production occurs at higher NO concentrations.

Model studies indicate that chemical $O_3$ destruction is generally found in the lower troposphere over the oceans due to low NO and high $H_2O$ concentrations, while generally higher $NO_x$ concentrations in the continental boundary layer lead to net $O_3$ production (Klonecki and Levy, 1997). Over the oceans $O_3$ loss extends to the free troposphere, while enhanced $NO_x$ due to lightning, convective up-lift from anthropogenic or biomass burning sources and downward transport from the stratosphere leads to $O_3$ production in the free troposphere and tropopause region (Roelofs and Lelieveld, 1997). This difference between oceanic and continental free troposphere vanishes in the upper troposphere, where $O_3$ production prevails (Klonecki and Levy, 1997; von Kuhlmann et al., 2003). Studies that infer net ozone production at least in part from in-situ measurements are rare and often limited to the boundary layer (Ren et al., 2013; Liu et al., 2012; Sommariva et al., 2011; Kleinman et al., 2005; Fischer et al., 2003; Kanaya et al., 2002; Kleinman, 2000; Zanis et al., 2000; Penkett et al., 1997; Cantrell et al., 1996). A number of studies based on aircraft measurements have been performed, using in-situ $O_3$, CO, $NO_x$, volatile organic compounds (VOC) and radiation measurements in combination with a box model to calculate $HO_x$ and $RO_x$ radical levels to study NOPR in the free troposphere (Kuhn et al., 2010; Kondo et al., 2004; Davis et al., 2003; DiNunno et al., 2003; Ko et al., 2003; Reeves et al., 2002; Olson et al., 2001; Schultz et al., 1999; Crawford et al., 1997a; Crawford et al., 1997b; Davis et al., 1996; Jacob et al., 1996). NOPR studies based on in-situ $HO_x$ or $RO_x$ measurements by aircraft have been performed by Olson et al., 2012; Ren et al., 2008; Cantrell et al., 2003a and Cantrell et al., 2003b. Carzola and Brune (2010) described the application of an in-situ instrument to measure ozone production, while estimating the NOPR requires in-situ measurements of radicals (OH, $HO_2$, $RO_2$), nitrogen oxide (NO) and photolysis rates (i.e. $J(O^1D)$) in addition to ozone and water vapour. Here we present airborne in-situ measurements of radicals and ozone precursors over the tropical rainforest in South-America during the GABRIEL (Guyanas Atmosphere-Biosphere exchange and Radicals Intensive Experiment with a Learjet) campaign in October 2005, and compare with a series of north-south transects over Europe in the extratropical troposphere as part of the HOOVER (HOx OVer EuRope) campaign in 2006 and 2007. For the first time the NOPR over the tropical rainforest in South America as well as over Europe is evaluated based only on in-situ measurement data (except peroxy radicals) and compared to a 3D chemical transport model.

## 2 Methods

### 2.1 GABRIEL and HOOVER measurements

The GABRIEL campaign took place in October 2005 over the tropical rainforest in French Guyana and Surinam. A total of 10 measurement flights, each between 3 and 3.5 hours long, were performed between 3° and 6°N and 59° to 51°W at altitudes between 300 and 9000 m (Figure 1a). All flights followed a similar pattern, with take-off from Zanderij airport

(Surinam, 5.3°N, 55.1°W), followed by a high altitude stretch east over the Atlantic Ocean, and a descent into the marine boundary layer off the east coast of South America. Turning west the aircraft followed the main wind direction in-land, performing flights in and out of the continental boundary layer over the rainforest. Finally, before landing at the home base a high altitude profile was flown over Surinam. Additionally, similar flight profiles were performed in the N-S direction. Take-off times of the flights were varied over the campaign in order to investigate diurnal variations. Details of the scientific objectives, measurement and model results can be found in Lelieveld et al. (2008) and the GABRIEL special issue in Atmospheric Chemistry and Physics (http://www.atmos-chem-phys.net/special_issue88.html).

HOOVER consisted of a total of two measurement campaigns in October 2006 and July 2007, composed of 4 measurement flights each. The measurements covered Europe from 40° to 75°N between 8° and 15°E and up to a maximum altitude of 12 km (Figure 1b). From the home base Hohn (Germany, 54.2°N, 9.3°E) regular research flights were performed southward with a stop-over at Bastia airport, Corsica (France, 42.2°N, 9.29°E) and northward with a stop-over at Kiruna airport (Sweden, 67.5°N, 20.2°E). The majority of the flights were performed in the upper troposphere, but regular profiles were flown in and out of the home and stop-over bases, as well as approximately half way towards the respective destination in either Southern Germany or Northern Scandinavia. Additional flights in summer 2007 were directed to the Arctic (Svalbard, Norway, 78.1°N, 15.3°E) and two flights over central Germany to study the influence of deep convection. Details about the campaigns can be found in two previous publications (Klippel et al., 2011; Regelin et al., 2013).

## 2.2 Observations

During both campaigns a Learjet 35A from GFD (Hohn, Germany) was used. This jet aircraft has a range of 4070 km and a maximum flight altitude of approximately 14 km. In the present configuration both the horizontal and vertical range were limited due to the use of two wing-pods housing additional instrumentation. The scientific instrumentation was similar during both campaigns. It consisted of a chemiluminescence detector (ECO Physics CLD 790 SR, Switzerland) for NO, $NO_2$ and $O_3$ measurements (Hosaynali Beygi et al., 2011), a set of up- and downward looking $2\pi$-steradian filter radiometers for $J(NO_2)$ measurements (Meteorologie Consult GmbH, Germany), a quantum cascade laser IR-absorption spectrometer for CO, $CH_4$ and HCHO measurements (Schiller et al., 2008), a dual enzyme fluorescence monitor (model AL2001 CA peroxide monitor, Aero-Laser GmbH, Germany) to measure $H_2O_2$ and organic hydroperoxides (Klippel et al., 2011), a laser induced fluorescence (LIF) instrument for simultaneous measurements of OH and $HO_2$ (Martinez et al., 2010; Regelin et al., 2013), a non-dispersive IR-absorption instrument (model LI-6262, Li-COR Inc., USA) for $CO_2$ and $H_2O$ measurements (Gurk et al., 2008), a proton transfer reaction mass spectrometer (PTR-MS, Ionicon, Austria) for partially oxidized volatile organic compounds measurements and a series of canisters for post flight analysis of non-methane hydrocarbons (Colomb et al., 2006). Here a sub-set ($O_3$, NO, CO, $CH_4$, $H_2O$, OH, $HO_2$ and $J(NO_2)$) of these measurements will be used to deduce NOPR values. Details about the performance of those measurements with respect to time resolution, precision, detection limits and total uncertainties can be found in Table 1.

## 2.3 Estimating peroxy radical concentrations and $J(O^1D)$

Most species that are needed for an evaluation of equations 1 – 4 are provided by in-situ observations with the exception of $[CH_3O_2]$ in Eq. 1 and $J(O^1D)$ in Eq. 2, that have to be derived from other measurements.

As mentioned in the introduction, we assume that in remote areas outside of the continental boundary layer, the concentrations of other volatile organic compounds besides methane are low, so that $CH_3O_2$ is the only organic peroxy radical at significant concentrations in view of $O_3$ formation. According to R1 and R2 the production rates for $HO_2$ and $CH_3O_2$ radicals are proportional to the concentrations of CO and $CH_4$, respectively. Since the photochemical lifetimes of both radicals with respect to their reaction with NO (R4, R5) or self-reactions leading to peroxides are similar (Hosaynali Beygi et al., 2011; Klippel et al., 2011), we assume that the ratio of $HO_2/CH_3O_2$ is proportional to their production rates $k_{(CO + OH)}[CO][OH]/k_{(CH4+OH)}[CH_4][OH]$, so that the concentration of $CH_3O_2$ can be deduced from equation 5:

$$[CH_3O_2] = \frac{k(2)[CH_4]}{k(1)[CO]} [HO_2] \tag{Eq. 7}$$

Thus using measured mixing ratios for CO, $CH_4$ and $HO_2$ and the temperature dependent rate coefficients for R1 and R2, the mixing ratio of $CH_3O_2$ can be estimated. Hosaynali Beygi et al. (2011) have used this approach in the marine boundary layer and compared it to both box model and 3D chemical transport model simulations in order to demonstrate the applicability of Eq. 7 in remote regions. We expect this to also hold for the free troposphere. In the continental boundary layer and in the outflow of deep convective clouds, R4 will most probably underestimate peroxy radical concentrations and thus $O_3$ production according to Eq. 1.

The $O_3$ photolysis rate $J(O^1D)$ and $J(NO_2)$ were calculated with the radiation transfer model TUV (https://www2.acom.ucar.edu/modeling/tropospheric-ultraviolet-and-visible-tuv-radiation-model) (Madronich and Flocke, 1999) and scaled to the observed $J(NO_2)$ values. The scaling accounts for the effects of clouds that are not simulated by the TUV model, in particular enhanced up-welling radiation when flying over larger cloud decks. This method is not ideal, since it does not take into account the wavelength dependency of either transmission or reflection by clouds. Shetter et al. (2003) indicate that the TUV simulation of $J(NO_2)$ and $J(O^1D)$ compared to observations are accurate to within 6 – 18 % and 6 – 11 %, respectively.

## 2.4 MATCH simulations

To compare the experimentally derived NOPR values with model simulations the 3D chemistry transport model MATCH-MPIC (Lawrence et al., 2003; von Kuhlmann et al., 2003) (hereinafter referred to simply as MATCH) has been used. The model is driven by meteorological data from the National Centre for Environmental Prediction (NCEP) Global Forecast System (GFS). The chemical scheme, including details of the non-methane hydrocarbon chemistry, is described in von

Kuhlmann et al. (2003). The model was run with a resolution of approximately 2.8° x 2.8° in the horizontal direction and includes 42 vertical σ-levels up to 2 hPa. Emissions from anthropogenic and natural sources are based on the Emission Database for Global Atmospheric Research EDGAR v3.2 (Olivier et al., 2002). The model has been used for chemical weather forecasting to guide the day-to-day flight planning during GABRIEL and HOOVER. Here we used post-campaign analysis simulations to produce virtual flights through the model along the actual aircraft trajectories, as done in Fischer et al. (2006). From the model results NOPR values derived from a full chemistry scheme including also higher order peroxy radicals have been calculated for every point along the flight tracks.

## 3 Results

### 3.1 Data Processing

In the following, net ozone production rates in ppbv/h are calculated from in-situ data according to Eq. 4, with ozone production $P(O_3)$ calculated from Eq. 1, including Eq. 7 for $[CH_3O_2]$ and ozone loss $L(O_3)$ from Eq. 2 with Eq. 3 for α. A filter ($O_3 < 100$ ppbv) was applied to the data to exclude direct stratospheric influence. Data from two flights dedicated to the investigation of deep convection (GABRIEL flight GAB 08 on 12 October 2005 (Bozem et al., 2014) and HOOVER II flight 07 on 19 July 2007) were not included, and will be discussed separately. For the remaining data tropospheric NOPR rates were calculated along the flight tracks using merged data sets with a time resolution of 30 s. Instead of presenting the NOPR data as a time series for individual flights, we make use of the sampling strategy followed in the two campaigns. As can be seen in Figure 1, flights during GABRIEL were mostly oriented from east to west (in Fig. 1a), while flights during HOOVER had a north south orientation (Fig. 1b). Therefore, all NOPR data from the individual flights have been binned into altitude-longitude (GABRIEL) and altitude-latitude (HOOVER) bins. The bin size is 1 km in altitude, 0.5° in longitude (GABRIEL) and 2.5° in latitude (HOOVER). NOPR values are presented as median values for a given altitude/longitude (GABRIEL) or altitude/latitude (HOOVER) bin. Additionally, the 1σ-standard deviation of the individual NOPR values in the respective bin is given as a measure of the atmospheric variability. Please note, that median values are used throughout the manuscript for NOPR calculations instead of mean values, in order to limit the influence of extreme events. Such events mainly influence NOPR calculations at the highest and lowest altitudes, and are predominantly due to NO spikes associated with aircraft emissions in the proximity of the airports or in flight corridors. Since these events are rare and vary strongly in terms of NO enhancement, we do not filter the data, but instead use median values that are not affected by occasional peak values. The same applies to values below the detection limit (e.g. for radicals) that might otherwise bias the calculations. Differences between mean and median NOPR values are insignificant during GABRIEL and up to a factor of two in the continental boundary layer during HOOVER I as shown in the respective figures.

Since NOPR values can only be calculated for those bins that have at least one data point for each trace gas species needed in Eq. 1 – 7 ($O_3$, NO, CO, $CH_4$, $H_2O$, OH, $HO_2$ and $J(NO_2)$), missing data strongly limit data coverage. Data gaps during all

three campaigns are mainly due to the low duty cycle of the TRISTAR instrument used to sequentially measure HCHO, CO, and CH$_4$. Due to a longer time spent on measuring HCHO and regular HCHO background measurements, only 10 min per hour (16 %) were dedicated to the measurement of CO and CH$_4$. Additional data gaps during GABRIEL arose from a partial lack of H$_2$O measurements. To overcome this, altitude profiles (median and standard deviation) for CO, CH$_4$ and H$_2$O have been calculated for the GABRIEL data set to substitute missing values by median values from the profiles for a particular altitude bin. This way the available number of NOPR calculations could be increased by a factor of 4, without changing trends in NOPR for different regions. Similar data gaps for CO and CH$_4$ during HOOVER I and II have been handled accordingly. These data gaps mainly affect the calculation of CH$_3$O$_2$ radicals in Eq. 7.

Further, for HOOVER II, nitrogen oxide (NO) data are not available from the flights to and from Corsica, thus in particular south of the Alps NOPR calculations cannot be based on in-situ data. Based on an additional flight between the home base in Northern Germany and Northern Italy (HOOVER II flight 06 from Hohn to Baaden Airport (Germany) on 19 July 2007), an average NO profile has been calculated for the southern part of Europe and used as a proxy for missing NO values. Note that data from HOOVER II flight 07 on the same day have not been used, since they were affected by strong convection over south-eastern Germany. Uncertainties due to the missing NO data during HOOVER II will be discussed further below. We did not filter the data for the time of the day. All flights were performed during daylight hours between approx. 10:00 and 17:00 local time.

Simulation results for P(O$_3$), L(O$_3$) and NOPR along the flight trajectories obtained from MATCH have been filtered for stratospheric influence, processed and binned in a similar way as the in-situ data and will be presented together with the in-situ data.

## 3.2 NOPR for GABRIEL

Figure 2 shows results for NOPR calculations based on in-situ measurements (Fig. 2a) and MATCH simulations (Fig. 2c) for the GABRIEL campaign in October 2005. As mentioned above NOPR values have been calculated along the flight tracks and sampled into bins of 1 km height and 0.5° longitude. The median values of NOPR per bin are presented with different colours, ranging from blue (negative NOPR indicating net O$_3$ destruction) to red (positive NOPR indicating net O$_3$ production). The number of data points within a bin is given as a number in the lower left corner. The circle inside the bin is a measure of the variability, with a box filling circle indicating a variability of more than 50 % relative to the median. Note that a variability value is presented even if only 2 data points are available for an individual bin. Both figures are oriented from west to east, so that data over the Atlantic Ocean are on the right hand side of the figure. With the South American coastline located between 53.5°W and 53.0°W, bins east of this longitude are representative for marine air masses and bins towards the west represent continental air masses.

In the lowest bins (0 – 1 km altitude) representing the boundary layer, NOPR values indicate a change from O$_3$ destruction in the marine boundary layer (-0.2 to -0.4 ppbv/h between 51°W and 54°W) towards a highly variable O$_3$ production (0 – 0.6

ppbv/h) regime in the continental boundary layer over the tropical rainforest (54°W to 57.5°W) (Fig. 2a). Highest NOPR values are observed at the coast at 52.5°W – 53°W, due to local pollution enhancing NO (see the discussion of Fig. 3a further below) most probably in the vicinity of Cayenne, the capital of French Guyana. Note that the absolute values for NOPR in the boundary layer, in particular over land, are less reliable, since we do not consider the contributions of higher organic peroxy radicals to ozone production and also neglect an additional $O_3$ sink due to reaction with alkenes, in particular isoprene. In the free troposphere, above 1 km and below 6 km altitude, NOPR values are generally negative, with strongest $O_3$ destruction in the first 2 km above the boundary layer. Above 6 km there is again a region with slightly positive NOPR, hence net $O_3$ production. In general, above the boundary layer NOPR values exhibit no difference between marine and continental regions. Note that similar results are obtained from calculations based only on the sub-set of data points for which simultaneous in-situ measurements of all species necessary to calculate NOPR are available. Thus, replacing the missing values by median values from average profiles does not change the results significantly. This statement holds also for results from the other campaigns, with the exception of missing NO measurements south of 55°N during HOOVER II. This will be addressed in section 2.4.

The MATCH simulations (Fig. 2c) exhibit similar NOPR tendencies for the different altitude regimes, though the absolute values are generally smaller (-0.2 ppbv/h and 0.1 ppbv/h, respectively). Also, MATCH simulates net $O_3$ destruction in the continental boundary layer over the tropical rainforest, in contrast to the calculation derived from the in-situ observations (Fig. 2a).

To illustrate the differences in NOPR between observations and model simulations, average (median, mean, 1σ-standard deviation) altitude profiles for the individual production and destruction terms are given in Fig. 2b and d for the observations and the simulations, respectively. Production throughout the troposphere is dominated by the reaction of NO with $HO_2$ (red dots in Fig. 2b and d), while the reaction of NO with $CH_3O_2$ (red squares) is much smaller by about a factor of 2. This behaviour is seen in both the observations (Fig. 2b) and the model simulations (Fig. 2d), but the absolute values for the production rates differ by a factor of two, with the observations being higher than the simulations. The concentration of $HO_2$ (and thus according to Eq. 7 the concentration of $CH_3O_2$) decreases with altitude throughout the troposphere by roughly a factor of 10 (Kubistin et al., 2010) so that the increase in the production rates is mainly due to an increase in NO concentrations, partly due to the shift in the $NO_x$ partitioning at low temperatures (Brunner et al., 2001; Ziereis et al., 2000). Ozone destruction is dominated by photolysis up to an altitude of 6 km (blue dots in Fig. 2b and d) with much higher (up to a factor of 4) destruction rates deduced from the observations (Fig. 2b). The reactions between ozone and either $HO_2$ or $CH_3O_2$ (triangles) are rather constant throughout the troposphere with larger rates (factor of 2) derived from observations compared to the model simulations. It is worth mentioning that the destruction rates are proportional to the ozone concentration. Increasing mixing ratios with altitude thus compensate for the pressure drop leading to almost constant $O_3$ concentrations.

In Figure 3 the ratio between NO and $NO_{th}$ calculated from Eq. 7 is plotted for in-situ data (Fig. 3a) and MATCH simulations (Fig. 3d). According to Eq. 7 the measurement-calculated threshold NO concentration in the boundary layer is 9 pptv, while it increases to about 20 pptv above the boundary layer (Fig. 3b). This is mainly due to the decrease of observed

HO$_2$ and estimated CH$_3$O$_2$ concentrations above the boundary layer, which leads to an increase of the threshold NO value. Measured NO mixing ratios are higher than the threshold values in the continental boundary layer (approx. 2 times larger), and at altitudes above 6 km (up to 3 times larger) (Fig. 3a), indicating net ozone production regimes as shown by Fig. 2a.

The evaluation of MATCH shows a slightly different altitude-dependent behaviour of NO$_{th}$, with highest values (21 pptv) in the boundary layer, decreasing almost linearly with altitude to lowest values of 10 pptv at 8 km (Fig. 3d). This behaviour is due to the underestimation of the reaction of O$^1$D with H$_2$O, most likely due to an underestimation of lower tropospheric H$_2$O concentrations. Figure 3b shows that simulated NO concentrations in MATCH are almost always lower than the threshold values (NO/NO$_{th}$ ratio between 0 and 1), except at the highest altitudes (NO about 50% higher than NO$_{th}$), thus explaining the overall negative NOPR values in Fig 2b. Therefore, the deviations between model simulations and in-situ data for NOPR are due to differences in the threshold NO levels (NO$_{th}$) and generally lower NO concentrations in the model simulations. Discrepancies in NO$_{th}$ by the MATCH model are possibly related to the non-methane hydrocarbon chemistry scheme, which may underestimate radical recycling under low NO$_x$ conditions and high isoprene (Kubistin et al., 2010; Taraborrelli et al., 2012).

In general, the charts of the NO to NO$_{th}$ ratio resemble the NOPR charts, with ratios larger than unity corresponding to net ozone production and ratios less than unity to ozone destruction. The values also scale quantitatively, illustrating the strong dependency of NOPR on NO mixing ratios. At the rather moderate NO levels in the free troposphere this relationship seems to be linear. This behaviour is also found in the data from the HOOVER campaigns.

## 3.3 NOPR for HOOVER I

Figure 4 shows results for NOPR calculations based on in-situ (Fig. 4a) and MATCH simulations (Fig. 4d) for the HOOVER I campaign in October 2006 over Europe. For this campaign the data for NOPR have been combined into 1 km altitude and 2.5° latitude bins. The majority of the data is obtained in the upper troposphere, while vertical profiles are restricted to take-offs and landings in airports on Corsica (40° - 42.5° N), Hohn (52.5° - 55°N) and Kiruna (67.5° – 70°N). Additional profiles were flown north and south of the Alps (45° - 50°N) and over Sweden (60° - 62.5°N). Overall, the in-situ data indicate net ozone production (0.1 – 0.3 ppbv/h) throughout the troposphere, except at the most northern (0 ppbv/h) and southern (~ -0.1 ppbv/h) parts of the flights. Threshold NO values are between 15 and 20 pptv below 2 km altitude and approximately 10 pptv between 2 and 10 km. Above that altitude they linearly increase to approx. 35 pptv at 12 km (Fig. 5b). In regions with net ozone production (NOPR > 0 ppbv/h), measured NO concentrations are up to a factor of 4 higher than NO$_{th}$. In the regions with NOPR ≤ 0 ppbv/h the measured NO concentration is smaller than NO$_{th}$ (Fig. 5a).

MATCH simulations of NOPR (Fig. 4c) exhibit slightly lower values (0 – 0.1 ppbv/h) compared to those derived from in-situ data, but the general tendencies are reproduced well. Figure 4b and d indicate that this difference between observations and model simulations is mainly due to an underestimation of the NO plus HO$_2$ reaction (red dots) by the model (~ factor of 2). The other production term (NO + CH$_3$O$_2$) is similar for observations and model simulations (red squares). As has been

observed for GABRIEL the almost constant production terms are due to an increase of NO with altitude, while $HO_2$ and thus $CH_3O_2$ concentrations drop by approximately a factor of 6 (Regelin et al., 2013). Contrary to GABRIEL, ozone photolysis is not the dominant sink (blue squares in Fig. 4b and d), but is similar to the other destruction rates ($HO_2 + O_3$ and $OH + O_3$). The absolute destruction rates are comparable between observations and model simulations. This similarity of the destruction

rates in observations and model simulations is most probably responsible for the similarity of the $NO_{th}$ values. The altitude profile of $NO_{th}$ derived from observations and MATCH are very similar, with the absolute values below 10 km being only slightly different (8 pptv from MATCH compared to 10 pptv from the observations) (Fig. 5b and c). As indicated in Fig. 5a and c $NO/NO_{th}$ values are also comparable. The only exception is the vertical profile over Sweden, were the observations indicate strong ozone production due to high NO concentrations, which are not reproduced by MATCH. Overall, MATCH

tends to underestimate NO concentrations throughout the troposphere, possibly related to underestimated vertical mixing of pollution from the boundary layer or missing $NO_x$ reservoir species such as alkyl nitrates in the chemistry scheme.

### 3.4 NOPR for HOOVER II

As mentioned above, NOPR calculations for HOOVER II are strongly affected by the failure of the CLD instrument used for NO measurements on the flights to the south, from Hohn to Corsica and back. Figure 6a shows results for NOPR calculations based on this limited data set. The NOPR calculations based on in-situ data are limited to latitudes north of 50°N. At higher latitudes a similar pattern to that during HOOVER I is observed, with net ozone production in the boundary layer, negligible to negative NOPR in the middle troposphere and a tendency for moderate net ozone production in the upper

troposphere. This general pattern is reproduced by MATCH simulations north of 50°N as shown in Fig. 6d.

In order to improve the data coverage, in particular south of 50°N, we used an average NO profile measured on 19 July 2007. On this day, two measurement flights were performed (HOOVER II flights 06 and 07), to study deep convection over Southern Germany out of Baaden airport (48.4°N, 8.4°E). Since no nitric oxide measurements were obtained on the regular flights south, the transfer flight to Baaden airport was extended southward to Northern Italy. Thus a limited data set of NO

could be obtained south of 50°N. Profile information is available from a descent north of the Alps close to Oberpfaffenhofen (48.4°N, 11.1°E) and the landing at Baaden airport. From this data set an average profile was deduced and median values have been used in the calculation of NOPR south of 55°N (Fig. 6b). This led to overall negative ozone tendencies throughout the troposphere at latitudes south of 50°N in contrast to the MATCH simulations that predict net ozone production (Fig. 6d). Both the model and $NO_{th}$ calculations based on in-situ data indicate a threshold NO concentration of 15 – 20 pptv between

the boundary layer and approx. 9 km altitude (with strongly increasing $NO_{th}$ above this altitude), with the model calculating slightly smaller values. The MATCH model simulates NO concentrations south of 55°N that are a factor of 2 to 3 higher than the simulated $NO_{th}$. The NO values from the HOOVER II flight 06 profile used to fill in data gaps south of 50°N are more than 50 % lower than the deduced $NO_{th}$ based on in-situ observations. Thus the negative ozone tendencies in Fig. 5b

are mainly due to an underestimation of NO mixing ratios. Therefore, a sensitivity study was performed by doubling the NO concentrations obtained from the HOOVER II flight 06 profile. The results for NOPR are shown in Figure 6c. The doubling of NO mixing ratios leads to a shift to positive NOPRs south of 55°N and to a much better agreement with the model simulations shown in Fig. 5d. It should be mentioned that the enhanced NO mixing ratios are in rather good agreement with

NO measurements obtained over southern Germany and northern Italy in the summer of 2003 as part of the UTOPHIAN-ACT campaign (Stickler et al., 2006). Thus it seems that the NO mixing ratios from the HOOVER II flight 06 profile may not be representative of background NO south of 50°N. This sensitivity study clearly showed the dominant role of NO for the NOPR calculations for HOOVER II. Taking into account the uncertainty in the measured NO data south of 55°N it does not make sense to discuss and compare production and destruction terms as in GABRIEL and HOOVER I.

Overall, a general feature of the three missions is a tendency of net ozone production at the highest altitudes, although we calculate strong increases of $NO_{th}$ with altitude. For HOOVER II both model simulations and observation-based calculations indicate a doubling of $NO_{th}$ between 9 and 11 km altitude from approx. 20 pptv to more than 40 pptv. As mentioned in the previous sections, similar tendencies for $NO_{th}$ were also deduced for HOOVER I and GABRIEL. In order to maintain positive NOPR at high altitudes a strong enhancement of NO above 10 km is required. In the next section we will discuss the

influence of deep convection on NOPR based on two case studies during GABRIEL and HOOVER II, respectively.

## 3.5 The influence of deep convection on NOPR

As outlined in section 2.1 the analysis thus far has been restricted to "background conditions" by filtering data that have been affected by deep convection. Research flights to study the outflow of deep convection have been performed during both

GABRIEL (Bozem et al., 2014) and HOOVER II (Regelin et al., 2013). Details about the convective systems, the flight tracks and trace gas measurements can be found in these articles.

During GABRIEL the outflow of a single convective cell at 9 to 11 km showed enhancements relative to background mixing ratios for CO, NO, OH and $HO_2$ of 40%, 130%, 70% and 20%, respectively. The CO increase with altitude suggests a strong contribution of boundary layer air. Transport of $HO_x$ precursors from lower layers of the troposphere led to enhanced OH

and $HO_2$ concentrations in the outflow. The strong enhancement in NO is most likely due to additional NO production from lightning. Ozone mixing ratios were also enhanced (38%), which may contradict the expectation that transport of boundary layer air in convective systems should lead to a decrease of $O_3$ mixing ratios in the outflow relative to the undisturbed middle and upper troposphere. A detailed discussion of the trace gas budgets, in particular for $O_3$, can be found in Bozem et al. (2014). Figure 7a shows mean and median values for NOPR in the outflow at the 10.5 km bin relative to campaign

background median and mean altitude profiles for non-convective air masses. The median value of 0.2 ppbv/h (mean: 0.27 ± 0.13 ppbv/h) is roughly a factor of 3 higher than the background value (median: 0.06 ppbv/h; mean: 0.06 ± 0.04 ppbv/h). This is mainly due to the enhancement of NO, in addition to the increase of peroxy radical concentrations.

In order to compare the GABRIEL results for NOPR with literature data for the tropics we estimate daily $O_3$ production by multiplying the hourly value with a typical day length of 12 h, yielding a value of 2.4 ppbv/d (range: 1.62 – 4.77 ppbv/d). This is in good agreement with observations over the Brazilian rain forest during ABLE 2B (1.5 – 1.7 ppbv/d) for conditions with an inflow from rural areas (Pickering et al., 1992a). For storms with an inflow characterized by urban environments

generally higher values were deduced during ABLE 2B (16.5 – 17.2 ppbv/d) (Pickering et al., 1992a). Higher values were also observed for convective transport from biomass burning plumes during ABLE 2A over South America (7.4 – 8.5 ppbv/d) (Pickering et al., 1992b), while smaller values of the order of 1 – 1.5 ppbv/d have been observed over tropical oceans (Pickering et al., 1993; Schulz et al., 1999; Kita et al., 2003; Koike et al., 2007).

During HOOVER II an eastward moving mesoscale convective system developed over the southern part of Germany on July

19, 2007. During a research flight out of Baaden airport, in- and outflow of a strong convective cell were probed close to Dresden, the capital of the State of Saxony. The outflow at 10.5 km altitude showed enhancements relative to background mixing ratios for CO, NO, OH and $HO_2$ by 85%, 600%, 350% and 150%, respectively, due to almost undiluted transport of boundary layer air to the upper troposphere and strong lightning activity. Ozone in the outflow was 20% lower than in background air (Bozem et al., 2017). As shown in Figure 7b this leads to a median NOPR of 1.89 ppbv/h (mean: $1.9 \pm 0.28$

15    ppbv/h), a factor of 6 higher than the upper tropospheric background for HOOVER II (median: 0.29 ppbv/h; mean: $0.31 \pm 0.07$ ppbv/h). The NOPR in the case study over Europe is an order of magnitude larger than over South America during GABRIEL. Median mixing ratios for NO during HOOVER and GABRIEL were 0.96 ppbv and 0.1 ppbv, respectively. Thus the difference in NOPR for the two case studies is mainly due to different NO concentrations. The median daily net ozone production for the HOOVER case is 22.67 ppbv/d (range: 19.47 – 26.11 ppbv/d) and about a factor of 2 – 4 larger than

values reported in the literature for NH mid-latitudes, e.g. ~ 15 ppbv/d during PRESTORM, Oklahoma (Pickering et al., 1990; Pickering et al., 1992a), 10 – 13 ppbv/d during STERAO-A over North America (DeCaria et al., 2005), up to 5 ppbv during EULINOX over Europe (Ott et al., 2007), and 5 – 7 ppbv/d during DC3 over North America (Apel et al., 2015).

## 4. Discussion and conclusions

As mentioned in the introduction, observation based calculations of NOPR using airborne data are rare. The majority of these studies were made over the central and eastern Pacific (Cantrell et al., 2003a; Olson et al., 2001; Kondo et al., 2004; DiNunno et al., 2003; Davis et al., 2003; Schultz et al., 1999; Crawford et al., 1997a; Crawford et al., 1997b; Davis et al., 1996) or the Atlantic Ocean (Ren et al., 2008; Reeves et al., 2002; Jacob et al., 1996). Continental studies thus far are restricted to Australia (Ko et al., 2003), the east coast of North America (Ren et al., 2008), and high latitudes over north

America (Olson et al., 2012; Cantrell et al., 2003b). This study is the first performed over the rain forest in South America (GABRIEL) and over continental Europe (HOOVER). It is also the first study that compares observation based NOPR to a 3D model simulation. Previous publications have used constrained box models, which are optimal tools to test the ozone production mechanism.

The net $O_3$ tendencies derived from both in-situ observations and 3D-model simulations confirm earlier studies with net $O_3$ formation (NOPR > 0) taking place in the continental boundary layer and the upper troposphere (above approx. 7 km in the tropics and mid-latitudes), and net $O_3$ destruction (NOPR < 0) in the marine boundary layer and the lower free troposphere (between 1 and 6 km altitude). The main reason that explains this distinction is shown to be the NO concentration. Both

observations and model simulations indicate that the fate of ozone depends on the amount of NO relative to the threshold NO concentration, derived from Eq. 7. In our study, the observed NO concentrations are always close to $NO_{th}$ (between 10% and several 100%). The NOPR values are therefore almost linearly dependent on NO, being typical for $NO_x$ limited $O_3$ production regimes (Seinfeld and Pandis, 1998).

This strong NO dependency also affects the comparison between in-situ observations and model simulations. Although

NOPR values show similar tendencies, absolute values are often slightly different, with the modelled absolute NOPR values typically being smaller in magnitude than observed (both for production and destruction regimes). This is partly due to differences in measured and simulated NO concentrations, as has been discussed for HOOVER II. An additional factor influencing the comparison is the calculated threshold NO value, which often differs for the observations and model simulations. This indicates that the reasons for the model-observation differences are complex and depend on more than one

parameter.

Absolute values for NOPR are comparable to earlier observations. During INTEX-A, which was performed over the east coast of North America and the western Atlantic Ocean, mean NOPR values of 8.4 ppbv/day, -0.8 ppbv/day and 11.4 ppbv/day were observed for the boundary layer (BL), the middle troposphere (MT) and the upper troposphere (UT), respectively (Ren et al., 2008). Observed daily means of NOPR during HOOVER I (fall season) are ~ 2 ppbv/day (BL), zero

(MT) and ~ 1 ppbv/day (UT) increasing to ~ 4 ppbv/day (BL), -1 ppbv/day (MT) and ~ 1 ppbv/day (UT) during the summer measurements (HOOVER II). These values are close to observation based estimates by Olson et al. (2012) over North America during summer 2008 as part of the ARCTAS campaign (BL: 2 ppbv/day, MT: -1 ppbv/day, UT: 1 ppbv/day). Since no data have been reported previously for the troposphere over the tropical rainforest, absolute NOPR values for GABRIEL cannot be compared.

One remarkable feature is the shift to positive NOPR values above approx. 7 km altitude that is found in both observations and simulations. This shift occurs, although $NO_{th}$ shows a tendency to increase at the highest altitudes. In previous publications $NO_{th}$ has been designated as either $NO_{critical}$ (e.g. Cantrell et al., 2003a) or as NO compensation point ($NO_{comp}$) (Reeves et al., 2002). Cantrell et al. (2003a) point out that for a number of campaigns critical NO levels generally encompass a "triangular" envelop bounded by about 5 pptv and a line between 25 pptv at the surface and 5 pptv at 12 km altitude

(Figure 10e in Cantrell et al., 2003a). Both observation-derived and model calculated $NO_{th}$ values from this study fit into the triangle defined by Cantrell et al. (2003a), but differ at the highest altitudes. While the campaigns listed by Cantrell et al. (2003a) show either constant values throughout the troposphere, or decreasing values with altitude, we find an increase in $NO_{th}$ at the highest flight levels during both HOOVER campaigns. According to Eq. 6 this increase in $NO_{th}$ can be either due to an increase in the $O_3$ sinks (photolysis and subsequent reaction with $H_2O$ or reaction with OH or $HO_2$), which are directly

and positively correlated to the ozone concentration, or a decrease in the $O_3$ source (concentrations of $HO_2$ and $CH_3O_2$). Due to the pressure decrease by a factor of 4 from the surface to 10 km altitude the concentrations of the radical precursors (CO, $CH_4$) also decrease by a factor of 4 which strongly reduces the concentrations of $HO_2$ and $CH_3O_2$ radicals and thus the denominator of Eq. 6. Ozone exhibits strongly increasing mixing ratios with altitude that compensate the pressure reduction effect, leading to an almost constant concentration throughout the troposphere. Overall this leads to a rather invariable $O_3$ loss rate throughout the troposphere even at the strongly decreasing $H_2O$ concentrations in the upper troposphere. Most likely the observed increase of $NO_{th}$ in the tropopause region during the HOOVER campaigns is due to a combination of strongly increasing ozone concentrations and decreasing radical ($HO_2$, $CH_3O_2$) concentrations. Reeves et al. (2002) report a similar increase in $NO_{comp}$ observed in the middle troposphere over the Atlantic (Fig. 5b of Reeves et al., 2002). Here $NO_{comp}$ increases at an altitude of 3 km due to a strong increase in ozone. To obtain net $O_3$ production above 7 km altitude, a strong increase in NO concentrations with altitude is required. This NO increase with altitude is partly due to a shift in the partitioning in $NO_x$, yielding higher NO concentrations at high altitude due to the temperature dependency of the NO + $O_3$ reaction, whose rate constant decreases with decreasing temperature (Ehhalt et al., 1992; Ziereis et al., 2000). Additional sources of NO associated with convection, lightning and downward transport from the stratosphere further enhance NO at high altitudes (Schumann and Huntrieser, 2007), resulting in the positive NOPR values above 7 km altitude obtained for all three campaigns. The case studies of enhanced NOPR values associated with convection and lightning produced $NO_x$ in section 3.5 thus indicate that the general increase of NOPR above 7 km altitude is most likely due to the integral effect of convection on the upper troposphere.

*Acknowledgements.* The authors would like to thank Uwe Parchatka, Rainer Königstedt, and Corinne Schiller for their help with the measurements. Also we would like to acknowledge the support from the GABRIEL and HOOVER teams, envisvcope GmbH (Frankfurt), and GFD (Gesellschaft für Zielflugdarstellung, Hohn).

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

Table 1: Precision, accuracy and detection limit of the in-situ measurements used to deduce NOPR.

|  | Precision (1σ) | Accuracy | Detection limit |
|---|---|---|---|
| $J(NO_2)$ | 1% | 10% | |
| $CO$ | < 1 % | 1 % | |
| $CH_4$ | < 1 % | 1 % | |
| $H_2O$ | | 5 % | 200 ppmv |
| $O_3$ | 4 % | 2 % | 2 ppbv |
| $NO$ | 7 % | 12 % | 5 pptv |
| $OH$ | 7 % | 18 % | 0.02 pptv |
| $HO_2$ | 1 % | 18 % | 0.07 pptv |

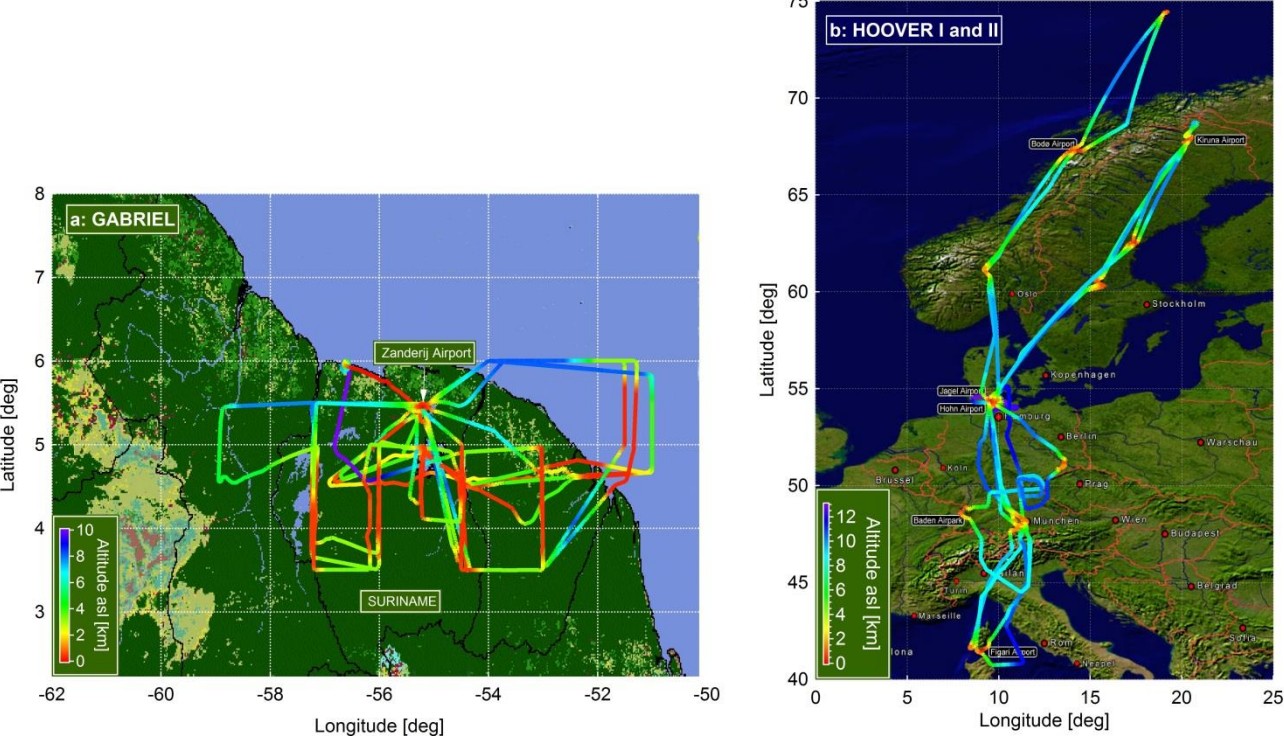

Figure 1: Flight tracks during GABRIEL (a: left panel) and HOOVER I and II (b: right panel).

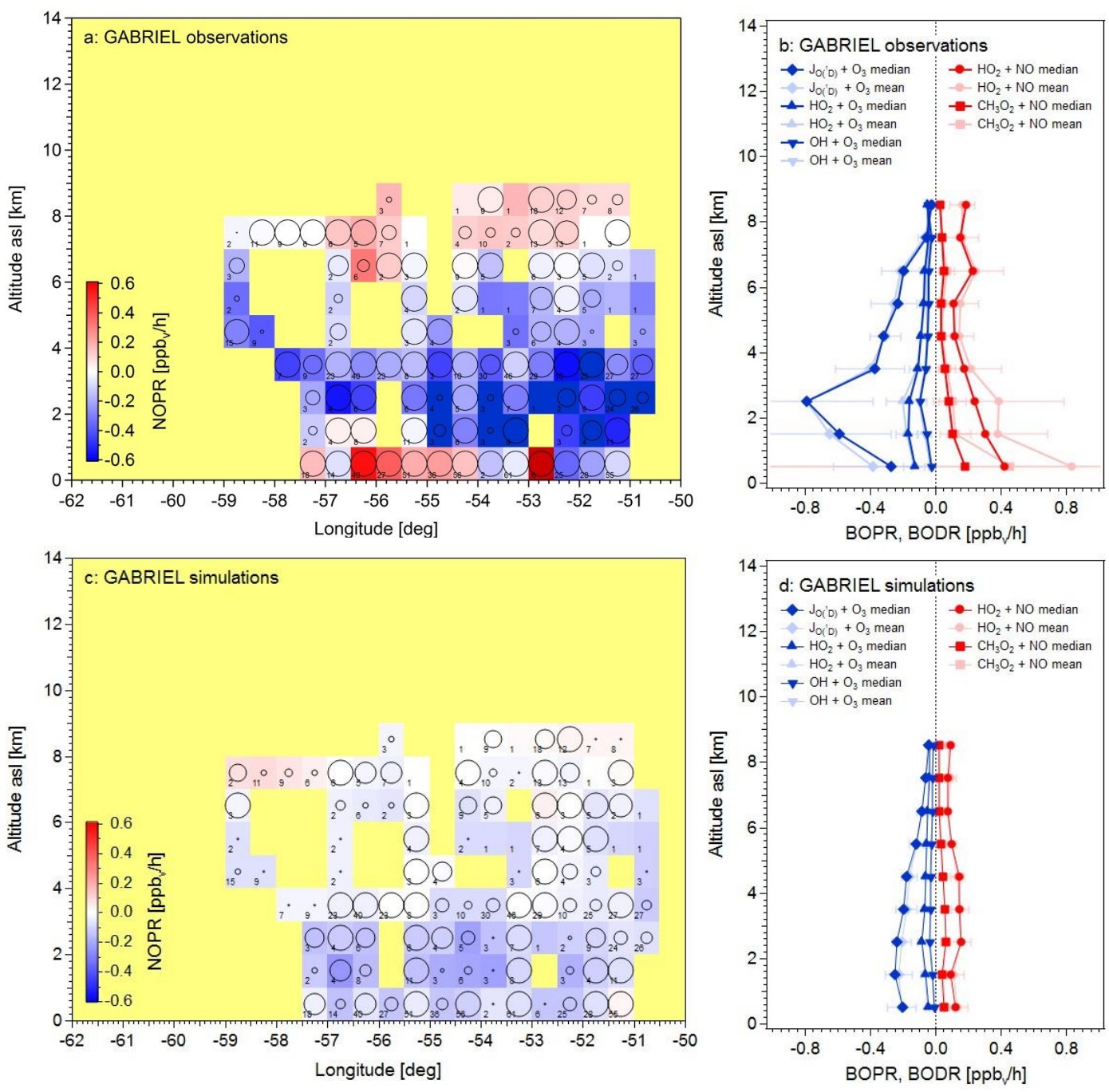

Figure 2: Net ozone production rates (NOPR) in ppbv/h calculated from in-situ data (a: upper panel) and from MATCH
5 simulations (c: lower panel) for GABRIEL. Altitude profiles of individual production and destruction rates are shown in b
(observations) and d (model simulations).

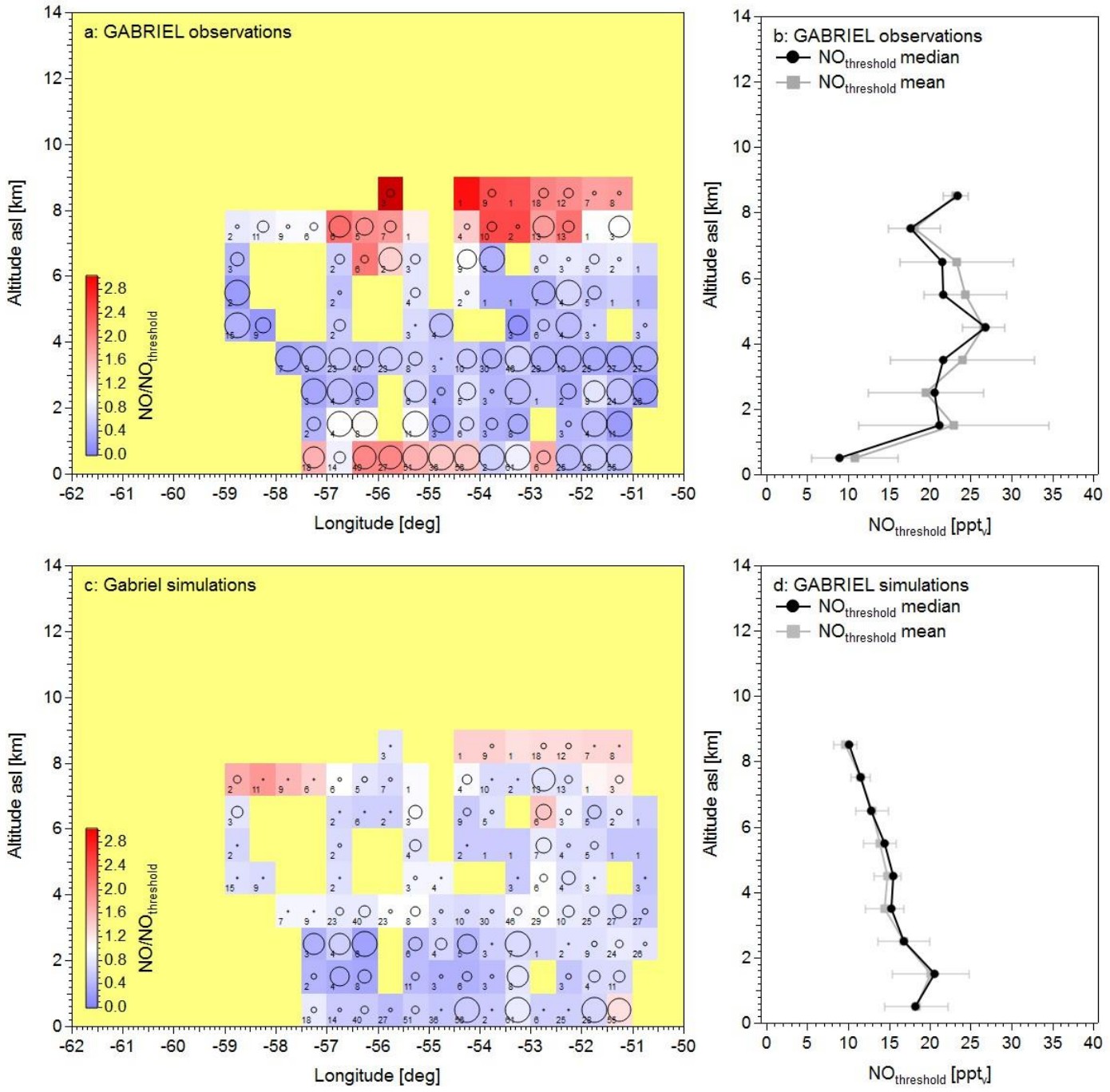

Figure 3: Ratio of NO to $NO_{th}$ deduced from in-situ data (a: upper panel) and MATCH simulations (c: lower panel) for GABRIEL. Altitude profiles for $NO_{th}$ are given in panel b for the observations and in panel d for model simulations.

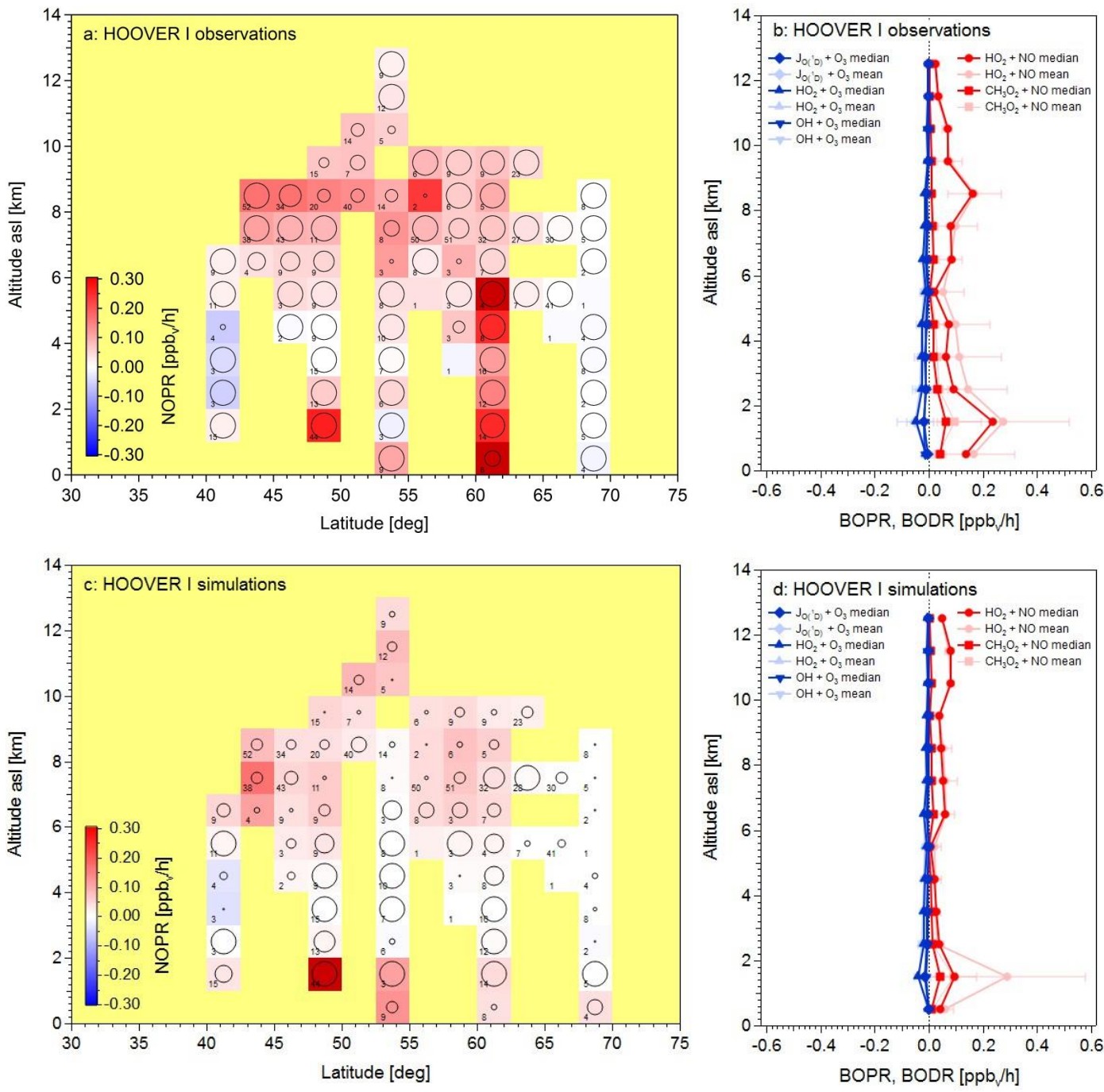

Figure 4: Net ozone production rates (NOPR) in ppbv/h calculated from in-situ data (a: upper panel) and from MATCH simulations (c: lower panel) for HOOVER I. Altitude profiles of individual production and destruction rates are shown in b (observations) and d (model simulations).

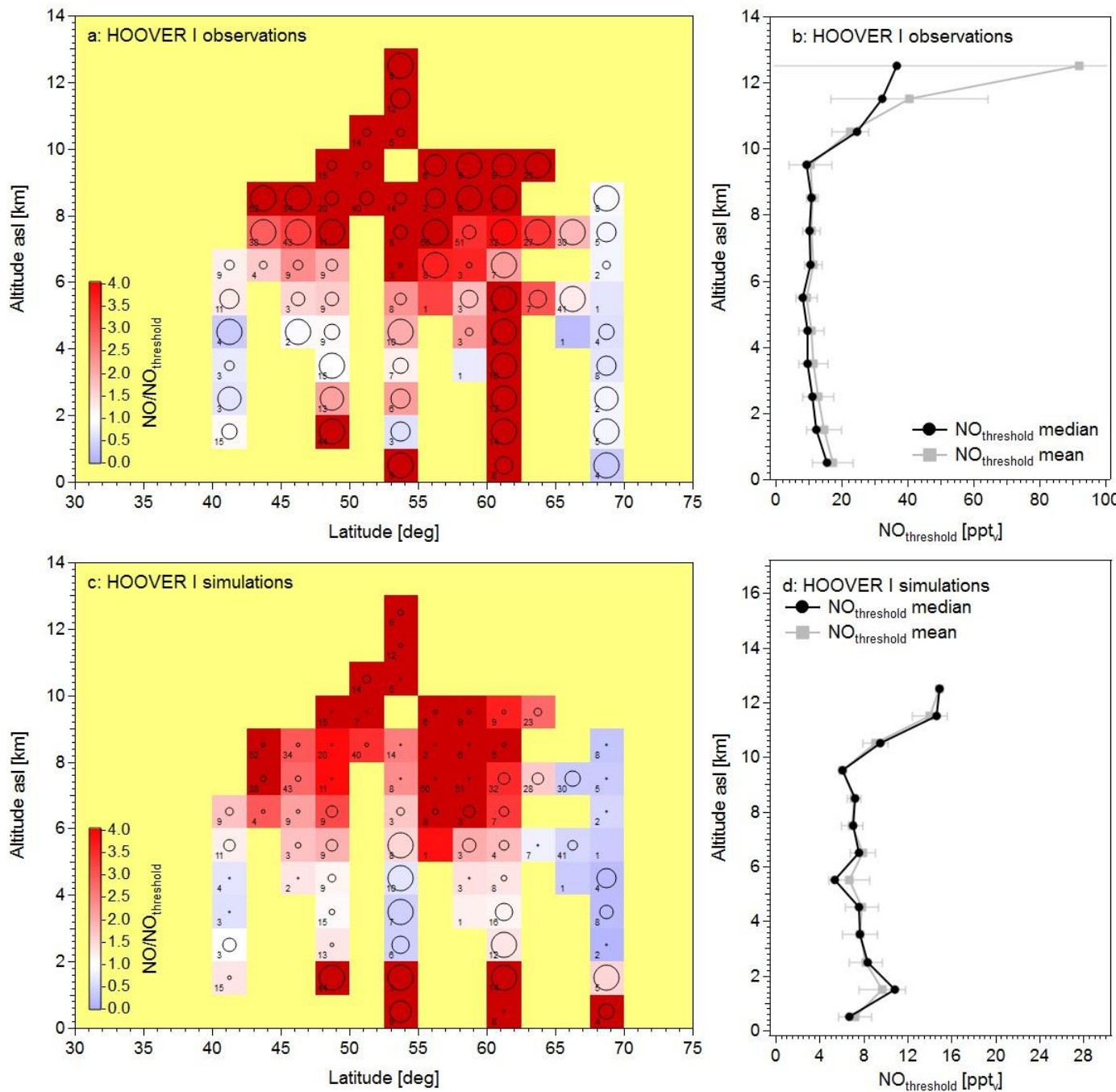

Figure 5: Ratio of NO to $NO_{th}$ deduced from in-situ data (a: upper panel) and MATCH simulations (c: lower panel) for
HOOVER I. Altitude profiles for $NO_{th}$ are given in panel b for the observations and in panel d for model simulations.

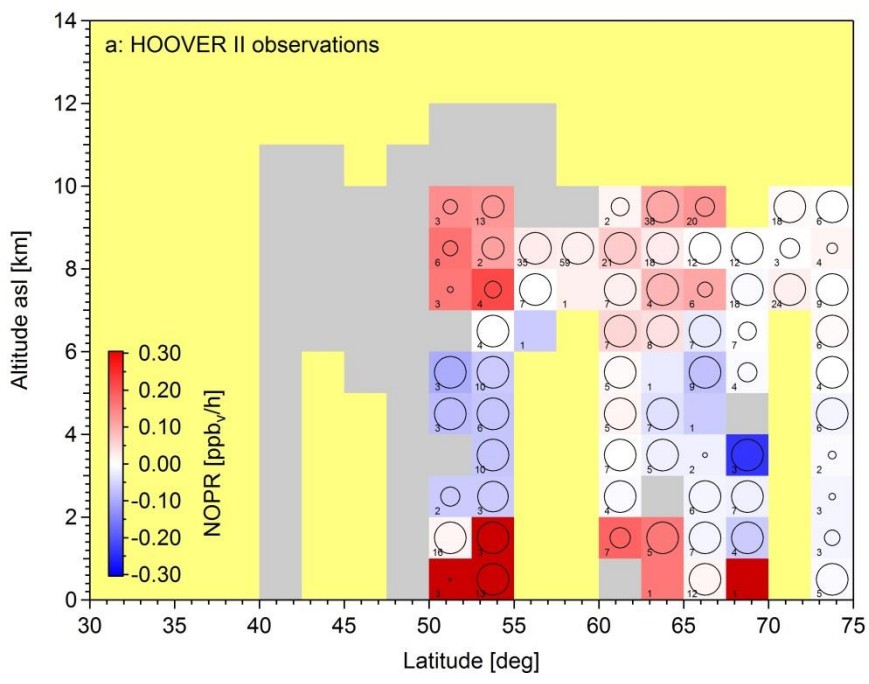

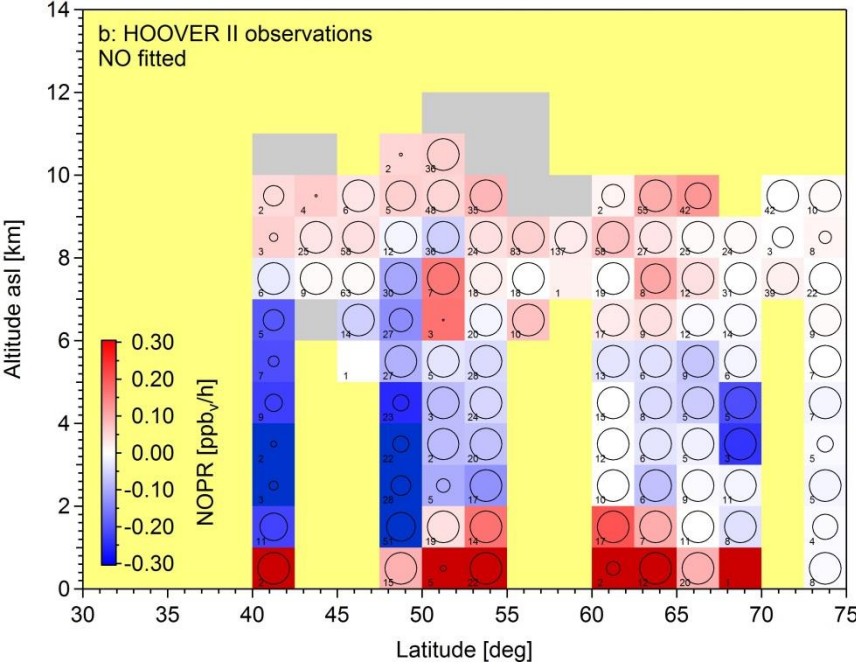

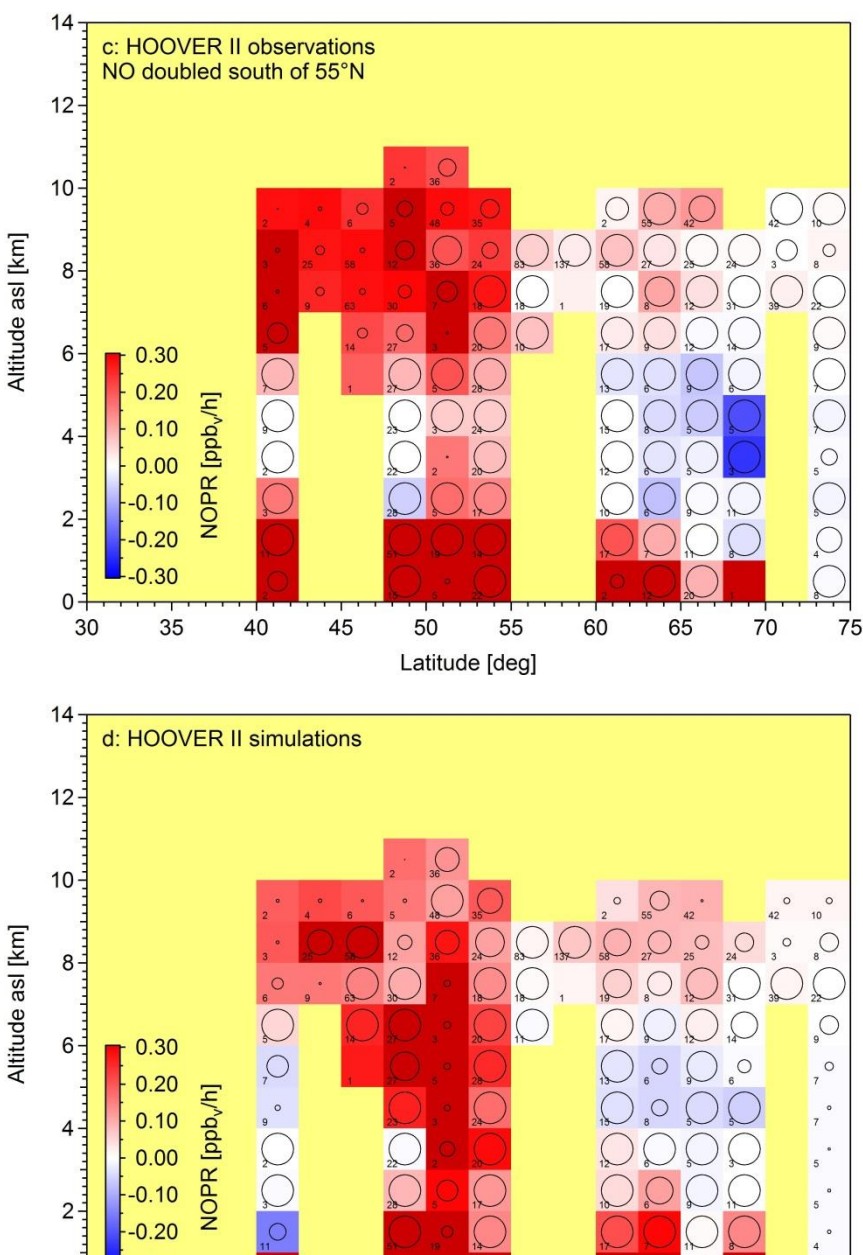

Figure 6: Net ozone production rates (NOPR) in ppbv/h calculated from in-situ data (a), fitted NO (b), double NO (c) and from MATCH simulations (b) for HOOVER II.

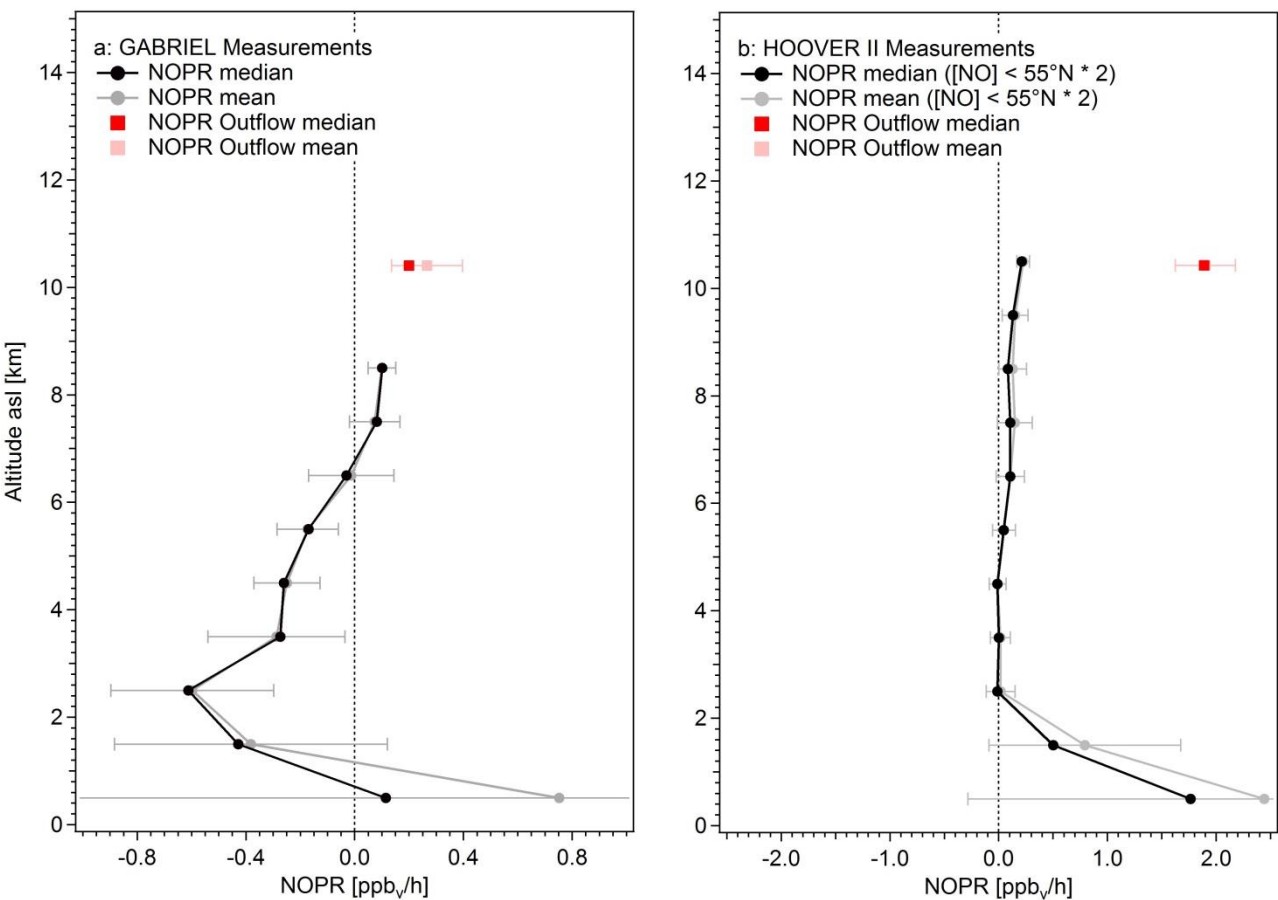

Figure 7: Net ozone production rates (NOPR) in ppbv/h (red squares) calculated from in-situ data for convective events during GABRIEL (a: left panel) and HOOVER II (b: right panel) relative to campaign average profiles (black).