# Peer review of "Chemical processes related to net ozone tendencies in the free troposphere"

_Atmospheric Chemistry and Physics, 2017_

## Referee Comment (RC1) · Anonymous Referee #1 · 22 Mar 2017

The submitted manuscript presents airborne in-situ measurements and model simulations of O3 and its precursors during tropical and extratropical field campaigns over South America and Europe aiming to calculate and assess the vertical distribution of net ozone production/destruction tendencies from both observations and model simulations. The manuscript has an added value on the understanding of the chemical control of ozone from the boundary layer to the upper troposphere over continental and marine environments in South America and Europe. I suggest acceptance of the manuscript for publication after taking into consideration the following comments.

Comments 1) page 3, lines 24-27: The authors cite a number of articles that infer net ozone production/destruction rates from in-situ observations (or at least in part) mentioning that the majority of these articles are limited to the boundary layer. I would suggest to distinguish which of these studies refer to the boundary layer and which to

the free troposphere. 2) page 5, lines 27-29: The authors calculate J(O1D) by scaling the TUV calculated J(O1D) using the ratio of observed J(NO2) and TUV calculated J(NO2). Are there any limitations in this method? If it is possible it would be nice if the authors could provide a reference providing some kind of evaluation of this scaling method. 3) Page 6, lines 20, 26 and 27: The authors use the acronym NOP instead of NOPR that use in the rest of the text. I would suggest to keep a consistency in the use of the acronym throughout the manuscript. 4) Page 6, line 32: The authors state that average altitude profiles for CH3O2 and H2O have been calculated for GABRIEL data. Do they mean CO instead of CH3O2 since the radical CH3O2 is then calculated from Eq.5? 5) Page 7, line 4: Could you please specify which exactly species have handled for data gaps in HOOVER I and II campaigns? 6) Page 8, lines 18-21: The authors discuss that the measurement-calculated threshold NO concentration increases from the boundary layer towards the free troposphere mainly due to the decrease of observed HO2 and estimated CH3O2 concentrations above the boundary layer. This could be further discussed if the authors consider that the NO threshold depends mainly to J(O1D), O3 and H2O and how these parameters vary from boundary layer to free troposphere. Of course the NO threshold depends also on other variables such as CO and CH4 concentrations, temperature and pressure. 7) Page 8, line 33: The authors mention that this behavior is also found in the data from the other campaigns. Which campaigns do they mean? HOOVER I and II? 8) Page 10, line 24: It is pointed that the analysis has restricted to background conditions by filtering data that have been affected by deep convection but there is no description somewhere in the manuscript how this filtering was done. 9) Discussion and conclusions: The NOPR values that have been calculated for the background conditions and presented in Sections 3.2, 3.3 and 3.4 should be also discussed in comparison with relevant calculations from other similar studies based on air-borne and in-situ observations.

---

## Referee Comment (RC2) · Anonymous Referee #2 · 20 Apr 2017

The paper offers an assessment of in situ ozone production rates based on observations from two field campaigns, GABRIEL and HOOVER. Ozone production rates derived from observations are compared to values obtained from a large scale chemical transport model, MATCH-MPIC. The analysis is somewhat limited by shortcomings in the observational dataset and the results confirm what has been well established by previous studies dating back more than two decades. The lack of novelty in the findings and quality of the dataset make it difficult to endorse publication. Specific comments and suggestions are offered below to expand on these points.

Comments on the Introduction:

This section of the paper fails to adequately recognize previous work and contains information that is both incorrect and incomplete that requires clarification and correction.

[Figure]

In trying to provide for some historical perspective, the authors provide a rather lean description of the relevant events and debate surrounding tropospheric ozone. Indeed, the reference to Junge regarding transport from the stratosphere and the 1960s references to LA are appropriate. However, it is key to note that a chemical explanation for tropospheric ozone was not available until Levy (1971) posited a source for OH in the troposphere and the development of a photochemical theory for tropospheric ozone was developed by Chameides and Walker (1973) and Crutzen (1973). It is also important to note that scientific debate on the relative importance of photochemistry versus downwelling from the stratosphere continued to be contentious for at least two more decades.

The capability for detailed ozone budget calculations by chemical transport models is indeed important, but this discussion is also unnecessarily limited. The major reference is to von Kuhlmann et al. (2003), but several more relevant and up-to-date assessments have occurred since then and should be recognized, e.g., Stevenson et al. (2006), Wu et al. (2007), and Wild et al. (2007). The range of values for budget terms provided from Kuhlmann et al. are based on a more limited sampling of models than from these other studies. It is also fails to recognize a couple of important aspects regarding the factors influencing ozone budget calculations in global models:

1) When discussing Net Ozone Production (NOP), the authors state that "The NOP itself is a delicate balance between two very large numbers. . ." referring to ozone production and destruction. This statement fails to recognize is that NOP has nothing to do with these larger terms in global model calculations. Instead, it is governed by the balance between ST exchange and surface deposition. Thus, when sampling across models, NOP is highly correlated to ST exchange (which tends to have the highest uncertainty) and is moderately correlated to surface deposition. By contrast, there is little correspondence between NOP and gross production and destruction terms across models.

2) There are VERY few models that infer net destruction of ozone globally, and these

are the models with very high estimates for ST exchange; thus, it is not incorrect to state that the vast majority of models calculate net ozone production. For example, in Stevenson et al. (2006) 20 of 21 models have positive NOP, in Wild et al. the few models with negative values are pre-2000 studies, and all models in Wu et al. exhibit positive NOP.

Discussion of net ozone production on page 3 (lines 12-17) is overly simplistic. Describing net ozone production as a "non-linear function of the concentration of peroxy radicals...as well as the concentration of NO" glosses over the subject in a way that does not provide any insight, especially given that there is no reference provided for a deeper discussion of this nonlinearity. More importantly, net production is not "non-linear" over much of the remote atmosphere since non-linearity is only present when there is enough NOx to influence the HOx budget to limit radical availability. As noted only a few lines above, you neglect the loss of NO2 due to reaction with OH, but it is precisely this reaction that often drives nonlinearity in ozone production. The discussion of threshold NO is also too simplified. A well-defined equation for this quantity is introduced later in the text, so why try to reduce it to competition between reactions R4 and R10? The rate constants for these two reactions have opposite temperature dependencies and R4 may be 4000 times faster than R10 near the surface, but this difference more than doubles at the colder temperature of the upper troposphere. Additionally, there are many environments where reaction R12a is the dominant ozone loss term rather than R4.

On page 3 (line 24) the authors state that "Studies that infer net ozone production at least in part from in-situ measurements are rare and often limited to the boundary layer..." I couldn't disagree more. The statement is followed by an extensive list of references (17 in all, with 11 focused only on the boundary layer), but this list of references overlooks a wealth of previous work that refutes this statement. The literature on ozone production assessed from in situ airborne measurements through the depth of the troposphere is prolific and covers many parts of the globe. I point the authors

to literature on North America in summer (Ren et al., 2008), the western North Pacific in different seasons (Davis et al., 1996 & 2003; Crawford et al., 1997a & 1997b), the South Atlantic (Jacob et al. 1996), the South Pacific in different seasons (Schultz et al., 1999; Olson et al. 2001), and the Arctic in different seasons (Olson et al., 2012). All of these references provide findings that corroborate the NOPR results shared in this paper, establishing that these features of ozone photochemistry have been well documented and understood for decades.

Comments on Data Processing:

While I appreciate the caveats presented by the authors, I have some concerns about the degree of inference used in the analysis of the observations.

Twice it is mentioned that median values are taken from average altitude profiles (bottom of page 6 and top of page 7). This does not make sense to me and needs to be clarified.

There is no discussion of filtering for time of day. What is the range of solar zenith angles for these measurements? Rather than calculate an average altitude profile for CH3O2 (page 6, line 32), wouldn't it be better to calculate an average CH3O2/HO2 ratio and scale CH3O2 to HO2? This would better capture variability in the photochemical environment which should affect CH3O2 and HO2 similarly.

When taking care of all data gaps, the authors increase the number of calculations for GABRIEL by a factor of 4 (page 7, line 3), but how can the reader be convinced that this leads to a more robust result? The number of calculations is increased "without changing trends in NOPR for different regions." This seems like a circular statement, since the expanded calculations rely heavily on inferences from the more limited dataset. If the trends don't change, then all of this extra effort seems of little value.

The use of an average NO profile for HOOVER calculations is even more disturbing given the critical role of NO in determining the strength of the ozone production rate. I

am not comfortable with this approach. Anyone experienced in airborne measurements will corroborate that NO is one of the most variable quantities in the atmosphere and that measurements from one day cannot be reasonably used to infer conditions on another day.

Comments on NOPR for GABRIEL:

Page 7, line29: The authors note that high NOPR at the coast is "probably due to local pollution in the vicinity of Cayenne." Looking at figure 2, this is one of the statistically weakest bins at the lowest altitude. So is this from a single flight through that box? Maybe twice? How representative then is this data point? You also have the data to back up the statement regarding pollution. Instead of guessing, you should corroborate the statement with some indication of the NO and CO levels seen in that box relative to the rest of the data set.

Page 7, lines 30-33: The reader is reminded that in the continental boundary layer, NOPR values are less reliable due to the inadequacy of equation 1. Ozone loss to reaction with isoprene is also mentioned, but should be much less important. Without any attempt to quantify this underestimation, it is difficult to place much value on these data. Why are you not taking advantage of the PTR-MS and canister data to at least put a semi-quantitative estimate on the likely influence of isoprene and other VOCs in the continental BL?

Page 8, lines 4-5: The authors state "Thus, replacing the missing values by median values from average profiles does not change the results significantly." This statement again indicates that the data filling process is somewhat circular, giving the illusion of a more robust result. There is no expectation of additional variance when using these median values to fill gaps. I also am still confused by "median values from average profiles".

Page 8, lines 7-10: The discussion of MATCH data in Figure 2a is inadequate. Which terms in equations 1 and 2 are responsible for these differences? Without deeper

discussion of the difference in precursors between the real atmosphere versus MATCH, it is hard to see why the effort was spent do the simulations.

Page 8, lines 16-18: The discussion of threshold NO should be expanded and related to earlier work. This quantity has been previously referred to as the "NO compensation point" or "critical NO" (see Reeves et al., 2002; Davis et al., 1996; Crawford et al., 1997; DiNunno et al., 2003; Kondo et al., 2004 and others). You will also notice that many of these references also refer to a critical NOx value that tends to have more predictable behavior. By comparing photochemistry at different altitudes for a given abundance of NOx, you eliminate the need to account for the large changes in partitioning between NO and NO2 that occur with altitude (and temperature).

Comments on NOPR for HOOVER I and HOOVER II:

Discussion of results for HOOVER I are cursory at best. A deeper discussion of the difference between the observations and MATCH is warranted.

The discussion for HOOVER II is slightly longer, but is dominated by treatment of the missing NO measurements for a portion of the flights. It is my opinion that these data should not be included as the attempt to salvage these runs comes with too much uncertainty.

Comments on Discussion and Conclusions:

As noted in the opening sentence, these results "confirm earlier studies". In that regard I struggle to find anything novel in the work and am dismayed by the level to which data gaps have had to be filled to get these results as compared to previous studies.

I continue to be concerned about the findings for threshold NO which is stated to have a "tendency to increase at the highest altitudes" (page 12, line 13). This is different than all previous studies and I am not convinced by the explanations offered. I have to take particular issue with the statement that "Overall this leads to a rather invariable O3 loss rate throughout the troposphere." It is well established that the ozone lifetime

increases with altitude by as much as an order of magnitude. This is mainly due to the dramatic decrease in water vapor which is both directly and indirectly responsible for ozone destruction. This also means that ozone destruction falls off more rapidly than production, which is only indirectly related to water vapor through radical availability. The amount of NO needed to overcome this disparity should decrease at the highest altitudes and is shown to do that in numerous studies (e.g., Reeves et al., 2002; Davis et al., 1996; Crawford et al., 1997; DiNunno et al., 2003; Kondo et al., 2004 and others).

In summary, the analysis presented is not sufficiently novel, lacks depth, and exhibits some behaviors that deviate from previous findings that do not seem plausible. Given the condition of the data set, I do not expect that these shortcomings can be overcome to generate findings worthy of publication.

Minor points:

Page 1, line 27: the use of "whose" inappropriately personifies O(1D). It would be better to rephrase as "...O(1D), which can subsequently react with water vapour to yield two OH radicals."

Page 2, line 2: The reference to von Kuhlmann et al., 2003 is for the wrong paper. These budget numbers come from the ozone manuscript, not the one on ozone-related species.

Reference list

Crawford, J., et al. (1997), An assessment of ozone photochemistry in the extratropical western North Pacific: Impact of continental outflow during the late winter/early spring, J. Geophys. Res., 102(D23), 28469–28487, doi:10.1029/97JD02600.

Crawford, J. H., et al. (1997), Implications of large scale shifts in tropospheric NO x levels in the remote tropical Pacific, J. Geophys. Res., 102(D23), 28447–28468, doi:10.1029/97JD00011.

Davis, D. D., et al. (1996), Assessment of ozone photochemistry in the western North

Pacific as inferred from PEM-West A observations during the fall 1991, J. Geophys. Res., 101(D1), 2111–2134, doi:10.1029/95JD02755.

Davis, D. D., et al. (2003), An assessment of western North Pacific ozone photochemistry based on springtime observations from NASA's PEM-West B (1994) and TRACE-P (2001) field studies, J. Geophys. Res., 108, 8829, doi:10.1029/2002JD003232, D21.

Jacob, D. J., et al. (1996), Origin of ozone and NOx in the tropical troposphere: A photochemical analysis of aircraft observations over the South Atlantic basin, J. Geophys. Res., 101(D19), 24235–24250, doi:10.1029/96JD00336.

Olson, J. R., et al. (2001), Seasonal differences in the photochemistry of the South Pacific: A comparison of observations and model results from PEM-Tropics A and B, J. Geophys. Res., 106(D23), 32749–32766, doi:10.1029/2001JD900077.

Olson, J. R., Crawford, J. H., Brune, W., Mao, J., Ren, X., Fried, A., Anderson, B., Apel, E., Beaver, M., Blake, D., Chen, G., Crounse, J., Dibb, J., Diskin, G., Hall, S. R., Huey, L. G., Knapp, D., Richter, D., Riemer, D., Clair, J. St., Ullmann, K., Walega, J., Weibring, P., Weinheimer, A., Wennberg, P., and Wisthaler, A.: An analysis of fast photochemistry over high northern latitudes during spring and summer using in-situ observations from ARCTAS and TOPSE, Atmos. Chem. Phys., 12, 6799-6825, doi:10.5194/acp-12-6799-2012, 2012.

Ren, X., et al. (2008), HOx chemistry during INTEX-A 2004: Observation, model calculation, and comparison with previous studies, J. Geophys. Res., 113, D05310, doi:10.1029/2007JD009166.

Schultz, M. G., et al. (1999), On the origin of tropospheric ozone and NOx over the tropical South Pacific, J. Geophys. Res., 104(D5), 5829–5843, doi:10.1029/98JD02309. Stevenson, D. S., et al. (2006), Multimodel ensemble simulations of present-day and near-future tropospheric ozone, J. Geophys. Res., 111, D08301, doi:10.1029/2005JD006338.

Wild, O.: Modelling the global tropospheric ozone budget: exploring the variability in current models, Atmos. Chem. Phys., 7, 2643-2660, doi:10.5194/acp-7-2643-2007, 2007. Wu, S., L. J. Mickley, D. J. Jacob, J. A. Logan, R. M. Yantosca, and D. Rind (2007), Why are there large differences between models in global budgets of tropospheric ozone? J. Geophys. Res., 112, D05302, doi:10.1029/2006JD007801.

---

## Author Comment (AC1) · 6 Jun 2017

**We thank the referee for her/his comments, that we will address point by point in our reply.**

The submitted manuscript presents airborne in-situ measurements and model simulations of $O_3$ and its precursors during tropical and extratropical field campaigns over South America and Europe aiming to calculate and assess the vertical distribution of net ozone production/destruction tendencies from both observations and model simulations. The manuscript has an added value on the understanding of the chemical control of ozone from the boundary layer to the upper troposphere over continental and marine environments in South America and Europe. I suggest acceptance of the manuscript for publication after taking into consideration the following comments.

Comments 1) page 3, lines 24-27: The authors cite a number of articles that infer net ozone production/destruction rates from in-situ observations (or at least in part) mentioning that the majority of these articles are limited to the boundary layer. I would suggest to distinguish which of these studies refer to the boundary layer and which to the free troposphere.

**Answer:**

**In the revised version of the manuscript we will differentiate between ground-based and air-borne studies. Additionally we will identify those airborne studies that used in-situ observations of radicals ($HO_x$, $RO_x$) instead of those that use radical concentrations derived from constrained box model simulations.**

2) page 5, lines 27-29: The authors calculate $J(O^1D)$ by scaling the TUV calculated $J(O^1D)$ using the ratio of observed $J(NO_2)$ and TUV calculated $J(NO_2)$. Are there any limitations in this method? If it is possible it would be nice if the authors could provide a reference providing some kind of evaluation of this scaling method.

**Answer:**

**The scaling accounts for the effect of clouds that are not simulated by the TUV model, in particular enhanced up-welling radiation when flying over larger cloud decks. This method is not ideal, since it does not take into account the wavelength dependency of either cloud transmission or reflection. Shetter et al. (Comparison of airborne measured and calculated spectral actinic flux and derived photolysis frequencies during the PEM tropics B mission, JGR, 108, D2, 8234, doi:10.1029/2001JD001320, 2003) indicate that the TUV simulation of $J(NO_2)$ and $J(O^1D)$ compared to observations are accurate to within 6 – 18 % and 6 – 11 %, respectively.**

3) Page 6, lines 20, 26 and 27: The authors use the acronym NOP instead of NOPR that use in the rest of the text. I would suggest to keep a consistency in the use of the acronym throughout the manuscript.

**Answer:**

**We will use NOPR throughout the revised manuscript.**

4) Page 6, line 32: The authors state that average altitude profiles for $CH_3O_2$ and $H_2O$ have been calculated for GABRIEL data. Do they mean CO instead of $CH_3O_2$ since the radical $CH_3O_2$ is then calculated from Eq.5?

**Answer:**

**Page 6, line 32 should read:**

*"To overcome this, average altitude profiles for CO, $CH_4$ and $H_2O$ have been calculated for the GABRIEL data set."*

5) Page 7, line 4: Could you please specify which exactly species have handled for data gaps in HOOVER I and II campaigns?

**Answer:**

**Data gaps during all three campaigns are mainly due to the low duty cycle of the TRISTAR instrument used to sequentially measure HCHO, CO, and $CH_4$. Due to a longer time spent on measuring HCHO and regular HCHO background measurements, only 10 min per hour (16 %) were dedicated to the measurement of CO and $CH_4$. Additional data gaps during GABRIEL arose from a partial failure of the $H_2O$ measurements. During HOOVER II the NO measurement failed on the regular southbound flights. In the revised manuscript we will clarify this.**

6) Page 8, lines 18-21: The authors dis-cuss that the measurement-calculated threshold NO concentration increases from the boundary layer towards the free troposphere mainly due to the decrease of observed $HO_2$ and estimated $CH_3O_2$ concentrations above the boundary layer. This could be further discussed if the authors consider that the NO threshold depends mainly to $J(O^1D)$, $O_3$ and $H_2O$ and how these parameters vary from boundary layer to free tro-posphere. Of course the NO threshold depends also on other variables such as CO and $CH_4$ concentrations, temperature and pressure.

**Answer:**

**In the revised manuscript we will add vertical profiles of $NO_{th}$, $NO/NO_{th}$, $P(O_3)$ and $L(O_3)$ for the individual processes described in R4, R5, R9, R10, and R12 to discuss differences between observations and model simulations in greater detail.**

7) Page 8, line 33: The authors mention that this behavior is also found in the data from the other campaigns. Which campaigns do they mean? HOOVER I and II?

**Answer:**

**Yes the other campaigns are HOOVER I and II. As mentioned above we will add vertical profiles for $NO_{th}$ and the NO to $NO_{th}$ ratio for all the campaigns discussed in our study.**

8) Page 10, line 24: It is pointed that the analysis has restricted to background conditions by filtering data that have been affected by deep convection but there is no description somewhere in the manuscript how this filtering was done.

**Answer:**

**Actually we did not filter the data for deep convection. Two flights,**

**one during GABRIEL and a second one during HOOVER II were dedicated to study the outflow of convective clouds. Those flights were discussed separately.**

9) Discussion and conclusions: The NOPR values that have been calculated for the background conditions and presented in Sections 3.2, 3.3 and 3.4 should be also discussed in comparison with relevant calculations from other similar studies based on air-borne and in-situ observations.

**Answer:**

**In the revised manuscript we will add a paragraph comparing our results to previous studies.**

---

## Author Comment (AC2) · 6 Jun 2017

**We thank the referee for her/his comments, that we will address point by point in our reply**

The paper offers an assessment of in situ ozone production rates based on observations from two field campaigns, GABRIEL and HOOVER. Ozone production rates derived from observations are compared to values obtained from a large scale chemical transport model, MATCH-MPIC. The analysis is somewhat limited by shortcomings in the observational dataset and the results confirm what has been well established by previous studies dating back more than two decades. The lack of novelty in the findings and quality of the dataset make it difficult to endorse publication. Specific comments and suggestions are offered below to expand on these points.

**Answer:**

**We regret that the referee feels that our manuscript suffers from a lack of novelty and shortcomings in the observational data set which we believe is not justified:**

*Novelty:*

**The referee states that this study only confirms results obtained from studies dating back more than two decades, and she/he points out a number of studies performed during several large airborne campaigns. Among the studies cited by the referee (9 in total) only 3 use in-situ measurements of radicals (Olson et al., 2001; Ren et al., 2008; Olson et al., 2012) to deduce net ozone production rates (NOPR). All other studies cited by her/him and in our manuscript use precursors (peroxides, CO, NMHC) to deduce $HO_x$ and $RO_x$ levels using box models. The studies that use in-situ observations of $HO_x$ radicals found significant differences between observations and constrained box model calculations for $HO_x$ in particular in the upper troposphere (e.g. Ren et al. state a median (mean) deviation between observed and modeled $HO_2$ at 8-12 km of 2.05 (5.49) (their Table 1) while Olsen et al. (2012) found median ratios (model/observation) for $HO_2$ of 0.49 for ARCTAS-A (their Table 5) and 0.60 for ARCTAS-B (their Table 6). This underscores that there is uncertainty in the $HO_x$ budget of the troposphere, in particular the UT, which was not detected before measurements of $HO_x$ on airborne platforms were introduced in the early 2000s. $HO_x$ plays a**

central role in NOPR (Eq. 4 in our manuscript) and threshold NO levels (Eq. 7), thus it can be expected that all previous studies suffered from this uncertainty. We agree with the referee, that the chemical mechanism for net ozone formation in the troposphere is well established, but in order to calculate NOPR and $NO_{th}$ one has to know the concentrations of all species affecting these calculations in great detail, either by in-situ measurements or modeling. Our study is, to the best of our best knowledge, only the fourth that uses in-situ observations of all relevant species (except peroxy radicals) to evaluate NOPR and the first over Europe and the rain forest in South America, respectively. From this perspective we consider our study to present novel results.

This is also the first study that compares observation based NOPR calculations (at least including $HO_x$ measurements) to simulations from a 3D chemical transport model. All previous studies cited by the referee or in our manuscript used constrained box models for comparison. Box models are obviously ideal tools to study a chemical mechanism and are thus adequate to study NOPR from observational data, but ozone budgets rely on 3D models and we believe that comparing with such a model adds a new dimension, since it provides additional information about how well a CTM or GCM simulates the 4 dimensional distribution of $O_3$ precursors and how this affects the model's capability to simulate $O_3$ distributions.

*Shortcomings in the observational data:*
We are sorry that we obviously did an inadequate job explaining how we dealt with missing data. Missing data arise mainly from different time resolutions and duty cycles. In order to calculate NOPR, simultaneous measurements of all species used in the calculation have to be available. $O_3$, $NO_x$, $HO_x$, water vapor and radiation are measured at 1 Hz resolution with a duty cycle of nearly 100 % (except during calibrations and background measurements). CO and $CH_4$ are measured together with HCHO by the three laser quantum cascade laser spectrometer TRISTAR in a time multiplexed mode (Schiller et al., 2008). Although this instrument also has a time resolution of 1 sec it measures species sequentially. Due to the low concentrations of HCHO the majority of the measurement cycle is dedicated to this species (60 %) leaving 20% each for CO and $CH_4$. Additional interruptions of ambient measurements are due to regular HCHO background measurements (20 – 50 s every 3-4 min) and calibrations every 30 – 40 min further reducing the duty cycle for CO and $CH_4$ to 16%, which we consider not to be a problem considering the relatively constant background concentrations.

Restricting calculations to only those times when all species are available would thus pose a significant limitation, in particular since CO and $CH_4$ are only used in Eq. 5 to calculate $CH_3O_2$. Additional

data gaps are due to instrument failures of TRISTAR and the $H_2O$ instrument on parts of the GABRIEL flights due to an overheated cabin. Instead of interpolation we used campaign averaged profiles (we will address the use of median instead of mean values later in our reply) for CO, $CH_4$ and $H_2O$ to fill in missing data during GABRIEL and HOOVER. This procedure was used since all species exhibited only small horizontal but large vertical variations. Together with measured $HO_2$ these average profiles were used to calculate $CH_3O_2$ radical concentrations.

To substantiate our hypothesis that this approximation (through average profiles) is sufficiently accurate we compared the reduced data set (only points when all species have been measured simultaneous) to an extended data set (factor 4 for GABRIEL) with added CO and $CH_4$ values and found that no significant difference for the calculated NOPR in a given bin. If our hypothesis would have been wrong (e.g. due to strong latitudinal gradients in CO, $CH_4$ or $H_2O$) one would expect to see some deviations. Since data gaps mainly affect the $CH_3O_2$ concentrations and our studies and those e.g. of Ren et al. (2008) indicate that ozone production due to the reaction of NO with $CH_3O_2$ is rather small at higher altitudes (less than 10% of the production due to NO + $HO_2$ above 6 km) it is not to be expected that our procedure to add missing data affects the results for NOPR at all.

Missing NO values are much more critical and our discussion in connection with HOOVER 2 clearly demonstrates that simple measures to infer NO concentrations from other flights or campaigns will most likely fail. Nevertheless, we would like to keep this discussion, since it nicely illustrates the sensitivity of NOPR to an accurate measurement of NO.

Comments on the Introduction:

This section of the paper fails to adequately recognize previous work and contains information that is both incorrect and incomplete that requires clarification and correction.

In trying to provide for some historical perspective, the authors provide a rather lean description of the relevant events and debate surrounding tropospheric ozone. Indeed, the reference to Junge regarding transport from the stratosphere and the 1960s references to LA are appropriate. However, it is key to note that a chemical explanation for tropospheric ozone was not available until Levy (1971) posited a source for OH in the troposphere and the development of a photochemical theory for tropospheric ozone was developed by Chameides and Walker (1973) and Crutzen (1973). It is also important to note that scientific debate on the relative importance of photochemistry versus downwelling from the stratosphere continued to be contentious for at least two more decades.

Answer:

We will follow the recommendation of the referee and will add a discussion of the critical role the studies of Levy, Chameides and Walker and Crutzen had on the development of the photochemical mechanism for ozone production in the troposphere. We will also

**indicate that the discussion on the role and strength of STE for the tropospheric Ozone budget is still not resolved.**

The capability for detailed ozone budget calculations by chemical transport models is indeed important, but this discussion is also unnecessarily limited. The major reference is to von Kuhlmann et al. (2003), but several more relevant and up-to-date assessments have occurred since then and should be recognized, e.g., Stevenson et al. (2006), Wu et al. (2007), and Wild et al. (2007). The range of values for budget terms provided from Kuhlmann et al. are based on a more limited sampling of models than from these other studies. It is also fails to recognize a couple of important aspects regarding the factors influencing ozone budget calculations in global models:

**Answer:**

**We will add these references. Originally we restricted our discussion to the von Kuhlmann paper since it describes results from the same model that we later use for the intercomparison.**

1) When discussing Net Ozone Production (NOP), the authors state that "The NOP itself is a delicate balance between two very large numbers. . ." referring to ozone production and destruction. This statement fails to recognize is that NOP has nothing to do with these larger terms in global model calculations. Instead, it is governed by the balance between ST exchange and surface deposition. Thus, when sampling across models, NOP is highly correlated to ST exchange (which tends to have the highest uncertainty) and is moderately correlated to surface deposition. By contrast, there is little correspondence between NOP and gross production and destruction terms across models.

**Answer:**

**The referee argues for a perspective of dominant processes which is not certain to apply to atmospheric models. Gross production and destruction in a global model is a summation over simulated $O_3$ production and destruction based on the model's chemical mechanism, emissions of precursors and their subsequent distribution due to transport. Stratosphere-troposphere transport of $O_3$ depends on the gradient of $O_3$ between the lower stratosphere and upper troposphere, which in turn both depends on and influences the photochemistry especially in the upper troposphere. So one can claim that either is the dominant process, and in this sense it can be claimed that ST exchange is adapted to NOP, and not necessarily vice versa, indicating that uncertainties in the models NOP force ST. Furthermore, in CTMs and GCMs the stratospheric source of $O_3$ is often highly parameterized, e.g. with prescribed $O_3$ concentrations in the lower stratosphere to reproduce measured ozone profiles. So we think it is fair to address the question whether the NOP in a model is accurately reflecting the processes in the atmosphere.**

2) There are VERY few models that infer net destruction of ozone globally, and these are the models with very high estimates for ST exchange; thus, it is not incorrect to state that the vast majority of models calculate net ozone production. For example, in Stevenson et al. (2006) 20 of 21 models have positive NOP, in Wild et al. the few models with negative values are pre-2000 studies, and all models in Wu et al. exhibit positive NOP.

Discussion of net ozone production on page 3 (lines 12-17) is overly simplistic. Describing net ozone production as a "non-linear function of the concentration of peroxy

radicals. . .as well as the concentration of NO" glosses over the subject in a way that does not provide any insight, especially given that there is no reference provided for a deeper discussion of this nonlinearity. More importantly, net production is not "non-linear" over much of the remote atmosphere since non-linearity is only present when there is enough $NO_x$ to influence the $HO_x$ budget to limit radical availability. As noted only a few lines above, you neglect the loss of $NO_2$ due to reaction with OH, but it is precisely this reaction that often drives nonlinearity in ozone production. The discussion of threshold NO is also too simplified. A well-defined equation for this quantity is introduced later in the text, so why try to reduce it to competition between reactions R4 and R10? The rate constants for these two reactions have opposite temperature dependencies and R4 may be 4000 times faster than R10 near the surface, but this difference more than doubles at the colder temperature of the upper troposphere. Additionally, there are many environments where reaction R12a is the dominant ozone loss term rather than R4.

**Answer:**

**We agree with the referee that our discussion of the nonlinearity of NOP is overly simplistic. We will replace this paragraph by:**

*"NOPR is nonlinear with respect to NO and peroxy radicals. This nonlinearity arises because $RO_x$ and $NO_x$ drive ozone production (R4-R6) but also terminate free radical chemistry (Puesede et al., 2015 doi: 10.1021/cr5006815):*

*$NO_2 + OH + M \rightarrow HNO_3 + M$* (R13)
*$NO_2 + RO_2 + M \rightarrow NO_2RO_2 + M$* (R14)
*$OH + HO_2 \rightarrow H_2O + O_2$* (R15)
*$HO_2 + HO_2 \rightarrow H_2O_2 + O_2$* (R16)
*$CH_3O_2 + HO_2 \rightarrow CH_3OOH + O_2$* (R17)

*Note that we neglect loss of $NO_2$ due to reaction R13 and R14 in Eq. 4. This is justified by the overall low $NO_x$ concentrations outside the continental boundary layer. Reactions R15 to R17 are also excluded since they affect $HO_x$ levels and would have to be taken into account to calculate their concentrations using a box model. Here we use observations of OH and $HO_2$ instead."*

**To address threshold NO we will move the presentation and discussion of Eq. 7 to the introduction and skip the discussion of competition between R4 and R10.**

3) On page 3 (line 24) the authors state that "Studies that infer net ozone production at least in part from in-situ measurements are rare and often limited to the boundary layer. . ." I couldn't disagree more. The statement is followed by an extensive list of references (17 in all, with 11 focused only on the boundary layer), but this list of references overlooks a wealth of previous work that refutes this statement. The literature on ozone production assessed from in situ airborne measurements through the depth of the troposphere is prolific and covers many parts of the globe. I point the author to literature on North America in summer (Ren et al., 2008), the western North Pacific in different seasons (Davis et al., 1996 & 2003; Crawford et al., 1997a & 1997b), the South Atlantic (Jacob et al. 1996), the South Pacific in different seasons (Schultz et al., 1999; Olson et al. 2001), and the Arctic in different seasons (Olson et al., 2012). All of these references provide findings that corroborate the NOPR results shared in this paper, establishing that these features of ozone photochemistry have been well documented and understood for decades.

**Answer:**
**We will add the above cited references to the paper and discuss ground-based and airborne studies separately. Here we will also emphasize that only a few airborne studies have been performed using in-situ observations of $HO_x$ and this is the first study performed for Europe and South America.**

Comments on Data Processing:

While I appreciate the caveats presented by the authors, I have some concerns about the degree of inference used in the analysis of the observations.

Twice it is mentioned that median values are taken from average altitude profiles (bottom of page 6 and top of page 7). This does not make sense to me and needs to be clarified.

**Answer:**
**Median values are used throughout the manuscript instead of mean values to limit the influence of extreme events. Such events mainly influence NOPR calculations at the highest and lowest altitudes and are predominantly due to NO spikes associated with aircraft emissions in the proximity of the airports or in flight corridors. Since these events are rare and vary strongly in the NO enhancement, we choose not to filter the data, but instead use median values that are not affected by a few high values. The same accounts for values below the detection limit (e.g. for radicals) that otherwise might bias the data. Differences between mean and median NOPR values are insignificant during GABRIEL and up to a factor of two in the continental boundary layer during HOOVER 1.**

**For consistency, we choose to use median instead of mean values for average CO and $CH_4$ profiles. Since these two species are hardly affected by extreme events (the only exception is a local fire in the boundary layer over Suriname during GABRIEL that was sampled on one flight yielding enhanced CO and $CH_4$ mixing ratios) the differences between profiles based on median and mean values is negligible.**

There is no discussion of filtering for time of day. What is the range of solar zenith angles for these measurements? Rather than calculate an average altitude profile for $CH_3O_2$ (page 6, line 32), wouldn't it be better to calculate an average $CH_3O_2/HO_2$ ratio and scale $CH_3O_2$ to $HO_2$? This would better capture variability in the photochemical environment which should affect $CH_3O_2$ and $HO_2$ similarly.

**Answer:**
**We did not filter the data for the time of the day. All flights were performed during daylight hours between approx. 10:00 and 17:00 local time.**

**Average profiles based on median mixing ratios for a given altitude bin were used to fill in data gaps in CO and $CH_4$ during all campaigns (and $H_2O$ in GABRIEL). Calculations of $CH_3O_2$ are based on Eq. 5 using actual measurements of $HO_2$ and the production rates. Page 6, line 32 should read:**

*"To overcome this, average altitude profiles for CO, CH₄ and H₂O have been calculated for the GABRIEL data set."*

When taking care of all data gaps, the authors increase the number of calculations for GABRIEL by a factor of 4 (page 7, line 3), but how can the reader be convinced that this leads to a more robust result? The number of calculations is increased "without changing trends in NOPR for different regions." This seems like a circular statement, since the expanded calculations rely heavily on inferences from the more limited dataset. If the trends don't change, then all of this extra effort seems of little value.

**Answer:**

**We explained the motivation for our procedure to fill in missing CO, CH₄ and H₂O data above. Using this procedure we ignore potential longitudinal (GABRIEL) or latitudinal (HOOVER) gradients in those species. To test the influence of this simplification by using only one average altitude profile we compared NOPR values calculated with and without the data gaps at various longitudes and latitudes. Since no differences were observed, we conclude that our hypothesis of a weak lateral dependency is correct. We cannot follow the referee in his statement that this method is circular. In the case of CO and CH₄ this might be fortuitous due to the small contribution of CH₃O₂ to NOP in the free troposphere as has been shown by Ren et al. (2008). To illustrate this, we will add vertical profiles of individual rates of ozone production and loss for all campaigns (observations and model results) to Fig. 2, 4 and 5.**

The use of an average NO profile for HOOVER calculations is even more disturbing given the critical role of NO in determining the strength of the ozone production rate. I am not comfortable with this approach. Anyone experienced in airborne measurements will corroborate that NO is one of the most variable quantities in the atmosphere and that measurements from one day cannot be reasonably used to infer conditions on another day.

**Answer:**

**As mentioned above we would like to keep this analysis in the paper to demonstrate exactly the point that the referee made: NO is most critical for NOPR and this is kind of a sensitivity study to demonstrate that even small errors or missing data for this central species have large consequences.**

Comments on NOPR for GABRIEL:

Page 7, line29: The authors note that high NOPR at the coast is "probably due to local pollution in the vicinity of Cayenne." Looking at figure 2, this is one of the statistically weakest bins at the lowest altitude. So is this from a single flight through that box? Maybe twice? How representative then is this data point? You also have the data to back up the statement regarding pollution. Instead of guessing, you should corroborate the statement with some indication of the NO and CO levels seen in that box relative to the rest of the data set.

**Answer:**

**The data in this bin is indeed obtained from a limited number of data points (6), indicating that not on all flights the crossing of the coastline has been made on the lowest level as can be seen from the**

data points at higher altitudes. The high NOPR in the bin is due to enhanced NO values as documented in Fig. 3a, with $NO/NO_{th}$ being enhanced by a factor 1.5, indicating that NO is at least a factor of 2 higher than in adjacent bins. This points to a local NO source (which is documented in Fig. 3a). So the "guessing" is only for Cayenne as the source of this local pollution. Therefore we reformulate this statement to:

*"…due to local pollution enhancing NO (see the discussion of Fig. 3a further below) most probably in the vicinity of Cayenne,…"*

Page 7, lines 30-33: The reader is reminded that in the continental boundary layer, NOPR values are less reliable due to the inadequacy of equation 1. Ozone loss to reaction with isoprene is also mentioned, but should be much less important. Without any attempt to quantify this underestimation, it is difficult to place much value on these data. Why are you not taking advantage of the PTR-MS and canister data to at least put a semi-quantitative estimate on the likely influence of isoprene and other VOCs in the continental BL?

**Answer:**

**We do not think that this can be easily done. Although it would be possible to estimate the amount of higher peroxy radicals from canister based NMHC measurements, one should keep in mind that this data set is rather limited since only 24 canisters were sampled per flight, so that the data coverage in the boundary layer is rather poor. Data coverage for isoprene is higher due to the PTRMS measurements but an estimation of its influence of NOPR is even more complex due to its dual role as a potential source of organic peroxides and as a sink for ozone due to the ozonolysis of isoprene. So we would like to maintain our caution about the limitations of our analysis in the boundary layer instead of speculating about the role of other peroxy radicals with the limited amount of data available.**

Page 8, lines 4-5: The authors state "Thus, replacing the missing values by median values from average profiles does not change the results significantly." This statement again indicates that the data filling process is somewhat circular, giving the illusion of a more robust result. There is no expectation of additional variance when using these median values to fill gaps. I also am still confused by "median values from average profiles".

**Answer:**

**See our comments above.**

Page 8, lines 7-10: The discussion of MATCH data in Figure 2a is inadequate. Which terms in equations 1 and 2 are responsible for these differences? Without deeper discussion of the difference in precursors between the real atmosphere versus MATCH, it is hard to see why the effort was spent do the simulations.

**Answer:**

**We agree with the referee that we could provide more details on the differences between observations and MATCH simulations with respect to NOPR, in particular since such a comparison has never been made before as mentioned above. To do so, we will extend**

**figures 2, 4 and 5 by adding average profiles of individual ozone production and destruction rates as well as the NOPR (similar to Figure 6) for observations and model result. Additionally, we will add profiles for NO$_{th}$ and the NO/NO$_{th}$ ratios for all campaigns, again for both observations and model data. This will allow us to address differences in precursor levels and their influence on NOPR.**

Page 8, lines 16-18: The discussion of threshold NO should be expanded and related to earlier work. This quantity has been previously referred to as the "NO compensation point" or "critical NO" (see Reeves et al., 2002; Davis et al., 1996; Crawford et al., 1997; DiNunno et al., 2003; Kondo et al., 2004 and others). You will also notice that many of these references also refer to a critical NO$_x$ value that tends to have more predictable behavior. By comparing photochemistry at different altitudes for a given abundance of NO$_x$, you eliminate the need to account for the large changes in partitioning between NO and NO$_2$ that occur with altitude (and temperature).

**Answer:**

**We will follow the referee's suggestion and compare our results to earlier studies. We are also aware that this quantity has been referred to as "NO compensation point" or "critical NO" in other studies. Nevertheless, we deliberately decided to call the quantity calculated in Eq. 7 a threshold value since it marks the change in a chemical regime, from ozone destruction to production. We don't believe that NO$_x$ (instead of NO) is a good indicator for this threshold, since it is NO that drives ozone production and our results indicate that there is some altitude dependency that might be masked by using NO$_x$ instead of NO.**

Comments on NOPR for HOOVER I and HOOVER II:

Discussion of results for HOOVER I are cursory at best. A deeper discussion of the difference between the observations and MATCH is warranted.

The discussion for HOOVER II is slightly longer, but is dominated by treatment of the missing NO measurements for a portion of the flights. It is my opinion that these data should not be included as the attempt to salvage these runs comes with too much uncertainty.

**Answer:**

**The discussions of results for HOOVER I and II will be extended in a similar way as discussed above for GABRIEL, in particular with respect to differences between observations and model results.**

Comments on Discussion and Conclusions:

As noted in the opening sentence, these results "confirm earlier studies". In that regard I struggle to find anything novel in the work and am dismayed by the level to which data gaps have had to be filled to get these results as compared to previous studies.

**Answer:**

**We have addressed these points above, early in our general reply to the referee.**

I continue to be concerned about the findings for threshold NO which is stated to have a "tendency to increase at the highest altitudes" (page 12, line 13). This is different than all

previous studies and I am not convinced by the explanations offered. I have to take particular issue with the statement that "Overall this leads to a rather invariable $O_3$ loss rate throughout the troposphere." It is well established that the ozone lifetime increases with altitude by as much as an order of magnitude. This is mainly due to the dramatic decrease in water vapor which is both directly and indirectly responsible for ozone destruction. This also means that ozone destruction falls off more rapidly than production, which is only indirectly related to water vapor through radical availability. The amount of NO needed to overcome this disparity should decrease at the highest altitudes and is shown to do that in numerous studies (e.g., Reeves et al., 2002; Davis et al., 1996; Crawford et al., 1997; DiNunno et al., 2003; Kondo et al., 2004 and others).

**Answer:**

**We agree with the referee that the ozone lifetime increases with altitude by almost an order of magnitude due to decreases in water vapor and slower reaction rates at lower temperatures. But this does not necessarily mean that the ozone destruction rate falls off faster than the production terms. The destruction term is proportional to the ozone concentration and increasing ozone mixing ratios (from approx. 20 ppbv close to the ground to around 100 ppbv close to the tropopause) will almost compensate the pressure drop by a factor of 5 (1000 hPa to 200 hPa). So in total the rate of ozone loss will probably decrease by an order of magnitude driven by the longer lifetime. This has to be compared to the change of $HO_2$ (and $CH_3O_2$) concentrations with altitude. Actually, if the drop in $HO_2$ concentrations with altitude is larger than the change in the total $O_3$ loss rate, Eq 7 predicts an increase in the $NO_{th}$ as observed in this study. Please note that such an increase might not be observed if one considers a threshold for $NO_x$ due to the change in partitioning.**

In summary, the analysis presented is not sufficiently novel, lacks depth, and exhibits some behaviors that deviate from previous findings that do not seem plausible. Given the condition of the data set, I do not expect that these shortcomings can be overcome to generate findings worthy of publication.

**Answer:**

**We hope that we have convinced the referee and editor that this paper holds enough novelty and sufficient data quality to revise this judgement of the manuscript.**

Minor points:

Page 1, line 27: the use of "whose" inappropriately personifies $O(^1D)$. It would be better to rephrase as ". . .$O(^1D)$, which can subsequently react with water vapour to yield two OH radicals."

Page 2, line 2: The reference to von Kuhlmann et al., 2003 is for the wrong paper. These budget numbers come from the ozone manuscript, not the one on ozone-related species.

**Answer:**

**We will address these points in the revised manuscript.**

**Reference list**

Crawford, J., et al. (1997), An assessment of ozone photochemistry in the extratropical western North Pacific: Impact of continental outflow during the late winter/early spring, J. Geophys. Res., 102(D23), 28469–28487, doi:10.1029/97JD02600.

Crawford, J. H., et al. (1997), Implications of large scale shifts in tropospheric NO x levels in the remote tropical Pacific, J. Geophys. Res., 102(D23), 28447–28468, doi:10.1029/97JD00011.

Davis, D. D., et al. (1996), Assessment of ozone photochemistry in the western North Pacific as inferred from PEM-West A observations during the fall 1991, J. Geophys. Res., 101(D1), 2111–2134, doi:10.1029/95JD02755.

Davis, D. D., et al. (2003), An assessment of western North Pacific ozone photochem-istry based on springtime observations from NASA's PEM-West B (1994) and TRACE-P (2001) field studies, J. Geophys. Res., 108, 8829, doi:10.1029/2002JD003232, D21.

Jacob, D. J., et al. (1996), Origin of ozone and NOx in the tropical troposphere: A pho-tochemical analysis of aircraft observations over the South Atlantic basin, J. Geophys. Res., 101(D19), 24235–24250, doi:10.1029/96JD00336.

Olson, J. R., et al. (2001), Seasonal differences in the photochemistry of the South Pacific: A comparison of observations and model results from PEM-Tropics A and B, J. Geophys. Res., 106(D23), 32749–32766, doi:10.1029/2001JD900077.

Olson, J. R., Crawford, J. H., Brune, W., Mao, J., Ren, X., Fried, A., Anderson, B., Apel, E., Beaver, M., Blake, D., Chen, G., Crounse, J., Dibb, J., Diskin, G., Hall, S. R., Huey, L. G., Knapp, D., Richter, D., Riemer, D., Clair, J. St., Ullmann, K., Walega, J., Weibring, P., Weinheimer, A., Wennberg, P., and Wisthaler, A.: An analysis of fast photochemistry over high northern latitudes during spring and summer using in-situ observations from ARCTAS and TOPSE, Atmos. Chem. Phys., 12, 6799-6825, doi:10.5194/acp-12-6799-2012, 2012.

Ren, X., et al. (2008), HOx chemistry during INTEX-A 2004: Observation, model calculation, and comparison with previous studies, J. Geophys. Res., 113, D05310, doi:10.1029/2007JD009166.

Schultz, M. G., et al. (1999), On the origin of tropospheric ozone and NOx over the trop-ical South Pacific, J. Geophys. Res., 104(D5), 5829–5843, doi:10.1029/98JD02309. Stevenson, D. S., et al. (2006), Multimodel ensemble simulations of present-day and near-future tropospheric ozone, J. Geophys. Res., 111, D08301, doi:10.1029/2005JD006338.

Wild, O.: Modelling the global tropospheric ozone budget: exploring the variability in current models, Atmos. Chem. Phys., 7, 2643-2660, doi:10.5194/acp-7-2643-2007, 2007. Wu, S., L. J. Mickley, D. J. Jacob, J. A. Logan, R. M. Yantosca, and D. Rind (2007), Why are there large differences between models in global budgets of tropo-spheric ozone? J. Geophys. Res., 112, D05302, doi:10.1029/2006JD007801.

**We would like to thank the referee for pointing out these additional references that we will include in our discussion.**